# Effects of undetected data quality issues on climatological analyses

Stefan Hunziker[1,2], Stefan Brönnimann[1,2], Juan Calle[3], Isabel Moreno[3], Marcos Andrade[3], Laura Ticona[3], Adrian Huerta[4], Waldo Lavado-Casimiro[4]

[1] Institute of Geography, University of Bern, Switzerland
[2] Oeschger Centre for Climate Change Research, University of Bern, Switzerland
[3] Laboratorio de Física de la Atmósfera, Instituto de Investigaciones Físicas, Universidad Mayor de San Andrés, La Paz, Bolivia
10 [4] Servicio Nacional de Meteorología e Hidrología del Perú (SENAMHI), Lima, Peru

Correspondence to: Stefan Hunziker (stefan.hunziker@giub.unibe.ch)

**Abstract.** Systematic data quality issues may occur at various stages of the data generation process. They may affect large fractions of observational datasets and remain largely undetected with standard data quality control. This study investigates the effects of such undetected data quality issues on the results of climatological analyses. For this purpose, we quality
20 controlled daily observations of manned weather stations from the Central Andean area with a standard and an enhanced approach. The climate variables analysed are minimum and maximum temperature, and precipitation. About 40 % of the observations are inappropriate for the calculation of monthly temperature means and precipitation sums due to data quality issues. These quality problems undetected with the standard quality control approach strongly affect climatological analyses, since they reduce the correlation coefficients of station pairs, deteriorate the performance of data homogenization methods,
25 increase the spread of individual station trends, and significantly bias regional temperature trends. Our findings indicate that undetected data quality issues are included in important and frequently used observational datasets, and hence may affect a high number of climatological studies. It is of utmost importance to apply comprehensive and adequate data quality control approaches on manned weather station records in order to avoid biased results and large uncertainties.

## 1 Introduction

Records of in situ weather observations are essential for climatological analyses. Although nowadays automatic stations are often in use, many national station networks have been based completely on manned station observations, and many still depend largely or partly on this type of observations. Various authors demonstrated possible errors in data records of manned stations (e.g. Rhines et al., 2015; Trewin, 2010; Viney and Bates, 2004). In order to detect and remove such errors, observational time series should be quality controlled before they are analysed (WMO, 2011, 2008). However, data quality issues are not always detected by common quality control (QC) methods (Hunziker et al., 2017). The overall impact of such undetected errors on climatological analyses is largely unknown.

In order to detect and remove non-climatic signals such as station relocations from observational data, station records should be homogenized (Aguilar et al., 2003; Brönnimann, 2015). For the success of the widely applied relative homogenization method, highly correlated time series are required (Cao and Yan, 2012; Gubler et al., 2017; Plummer et al., 2003; Trewin, 2013; Venema et al., 2012). Similarly, the important spatial consistency test in the QC process depends on suitable neighbouring stations (Durre et al., 2010; Plummer et al., 2003). Usually, the correlation between station pairs decreases with increasing distance. In some regions of the world, correlations are clearly lower or lose significance after shorter distances than in others (Gubler et al., 2017; New et al., 2000). According to Gubler et al. (2017), not only climatological factors may be responsible for such differences, but also factors related to the quality of the observations, such as station siting and observation practices. Besides potentially reducing the correlation between station pairs, data quality issues may also induce inhomogeneities in time series (WMO, 2008). As a result, the performance of statistical data homogenization methods is reduced due to the higher number of breakpoints (Domonkos, 2013). To the author's knowledge, the impact of data quality problems on station correlations and statistical data homogenization has not been thoroughly studied so far.

Trend magnitudes and signs in station records may strongly differ among neighbouring stations. This was observed in many parts of the world and for various climate variables and indices, such as minimum temperature (López-Moreno et al., 2016), precipitation (Rosas et al., 2016; Vuille et al., 2003), diurnal temperature range (Jaswal et al., 2016; New et al., 2006), or extremes indices (Skansi et al., 2013; You et al., 2013). Certain trend differences may be expected even on short spatial scales due to factors such as topography and feedback processes (You et al., 2010). However, errors in observations may also affect individual station trends and hence increase the trend spread within a region. Furthermore, regional trends may deviate from observations in comparable areas. For instance, studies found stronger positive trends in maximum than minimum temperatures since the middle of the 20[th] century in the Bolivian and Peruvian Altiplano (e.g. López-Moreno et al., 2016), and Alexander et al. (2006) detected a decrease in the number of warm nights in the same region. These findings are not in accordance with the globally observed and expected increase of night-time temperatures and decrease of the diurnal temperature range (Alexander et al., 2006; Donat et al., 2013b; IPCC, 2013; Morak et al., 2011; New et al., 2006; Quintana-Gomez, 1999; Vincent et al., 2005). Therefore, the question arises if non-climatic factors may cause systematic trend biases in entire regions.

The present study addresses the aforementioned research questions by applying two different QC approaches on the same observational dataset and comparing the results of relevant climatological analyses afterwards. As the standard QC approach, we used the method that is applied to the GHCN-Daily dataset (Menne et al., 2012). As the enhanced approach, we applied the QC tests suggested by Hunziker et al. (2017) that focus on the detection of systematically occurring data quality issues.

Since this is not a self-contained method, the GHCN-Daily QC was additionally applied afterwards.

The dataset used in the present study consists of manned station observations from the Central Andean region. This area is highly suitable for investigating the impacts of undetected data quality issues due to two main reasons: First, all the uncertainties discussed in the previous paragraphs are found in Central Andean station data, and second, data quality issues that may not be detected by standard QC methods are well studied (Hunziker et al., 2017). Furthermore, the topography in the

area is complex, and station density is sparse, making QC and data homogenization difficult. The dataset used contains the climatological key variables maximum temperature (TX), minimum temperature (TN), and precipitation (PRCP).

In this article, we first describe the data (Sect. 2) and explain the methods (Sect. 3). Next, we present the results (Sect. 4), in which we describe the frequency of the data quality issues (Sect. 4.1), and focus on their effects on the correlation of station pairs (Sect. 4.2), data homogenization (Sect. 4.3), and trends (Sect. 4.4).We discuss the results (Sect. 5), and finally draw the

conclusions of our findings (Sect. 6).

## 2 Data

The dataset used for the present study includes observational records from Bolivia (Servicio Nacional de Meteorología e Hidrología de Bolivia, and the Bolivian civil airport administration), from the Peruvian department of Puno (Servicio Nacional de Meteorología e Hidrología del Perú), and from some Chilean and Paraguayan stations located near the Bolivian boarder

(Dirección Meteorológica de Chile, Dirección de Meteorología e Hidrología (Paraguay)) (Fig. 1). The dataset was created within the framework of the project "Data on climate and Extreme weather for the Central AnDEs" (DECADE) and includes daily TX, TN, and PRCP measurements. All records in the DECADE dataset originate from manned weather stations. This reflects the conditions of weather observation networks in the Central Andean area, where only a few automatic weather stations are in service (Hunziker et al., 2017). The first records in the DECADE dataset date back to 1917, and the most recent

observations were taken in 2015. For more details on weather observations in the Central Andean region, see Hunziker et al. (2017).

The altitude of the stations in the study area ranges between 98 and 4667 m a.s.l. Stations at elevations ≤600 m a.s.l. group in the east (henceforward referred to as "Lowland stations"), while stations at elevations ≥3500 m a.s.l. are located in the west (henceforward referred to as "Altiplano stations") (see Fig. 1). Stations at altitudes in between the Lowlands and the Altiplano

group along the eastern slopes of the Central Andes (henceforward referred to as "Valley stations").

A large fraction of the 341 TX, 339 TN, and 698 PRCP time series in the original dataset cover only short observation periods or contain large gaps. Therefore, all records with a sum of measurements <20 years (i.e. <7300 valid observations) were

excluded. This nearly divided the number of time series in half, resulting in 180 remaining TX and TN, and 378 PRCP records. This dataset containing the raw data (i.e. not quality controlled or homogenized) is called "DATA$_{RAW}$" henceforward.

For the present study, the complete time series of DATA$_{RAW}$ were quality controlled and homogenized. However, for the subsequent analyses (i.e. error frequency, correlation, and trend analyses), only the period 1981 to 2010 was analysed. During

this 30 year standard period, the highest number of station records is available (104 TX, 106 TN, and 220 PRCP time series with ≥80 % of valid observations), and data quality is usually higher than earlier in time.

## 3 Methods

### 3.1 Quality control

DATA$_{RAW}$ was quality controlled with two different approaches. The first approach represents an established standard QC

method. Such methods mostly focus on the detection of single suspicious values (Hunziker et al., 2017). The second approach additionally takes systematically occurring data quality issues into account that may remain undetected with standard QC. It is therefore considered as enhanced QC.

#### 3.1.1 Standard approach

The Global Historical Climatology Network (GHCN)-Daily was developed for a wide range of applications, including studies

of extreme events (Menne et al., 2012), and it is the premier source of daily TX, TN, and PRCP observations from various regions of the globe (Donat et al., 2013a). The GHCN-Daily data are quality controlled with a comprehensive set of 19 QC tests, including spatial consistency tests (Durre et al., 2010). It is a fully automatic QC approach that was particularly developed to run unsupervised (Menne et al., 2012). Evaluations of the performance showed that the method is effective at detecting gross errors as well as more subtle inconsistencies though having a low false-positive rate (Durre et al., 2010). This QC method

was applied to DATA$_{RAW}$.

However, the detections (i.e. the flags for failing certain tests) of the GHCN-Daily QC had to be slightly adapted in order to be more appropriate for weather observations in the Central Andean region. One of the internal consistency tests detects cases where TX is lower than TN of the previous day. This test should guarantee the physical consistency of TX and TN measurements that are representative for a 24-h period. However, in various Bolivian stations (particularly stations at airports),

TX is representative for the afternoon hours only (observations start at noon and end in the evening). This measurement practice should avoid problems in attributing the observation to a specific calendar day. Usually, daily temperature maxima occur in the afternoon indeed. Nevertheless, during certain weather events (particularly the frequent cold surges in the Lowlands (e.g. Espinoza et al., 2013; Garreaud, 2001; Vera and Vigliarolo, 2000)), the temperature in the afternoon does not exceed the TN value measured in the morning. As a result, a high number of observations in the Lowlands was flagged. To the authors'

knowledge, this measurement practice has been applied to TX but not to TN observations, and no large scale changes of this practice in the Central Andean area are known. Therefore, this practice (even though not ideal) does not introduce any error or

bias as long as it remains unchanged. As a consequence, internal consistency flags set because of this particular QC test were regarded as invalid.

Furthermore, the GHCN-Daily QC did not flag a few extreme outliers. This may happen if a reported value exceeds the maximum of five places in tenth of °C or mm allowed in the GHCN-Daily data format (e.g. values $\leq$-10000). In order to

remove such erroneous numbers, we added an additional flag to all unflagged temperature values >70 °C and <-70 °C, as well as to all unflagged negative PRCP values.

In total, about 0.35 % (temperature) and 0.15 % (PRCP) of all measurements were flagged. This is similar to the overall fraction of 0.24 % flagged observations in the GHCN-Daily dataset (Durre et al., 2010). In DATA$_{RAW}$, about two-thirds of the flagged temperature and the great majority of the flagged PRCP observations are monthly or yearly duplicate data. For any

further analyses, all flagged values were removed. The dataset quality controlled with this standard QC approach will be called "DATA$_{QC-S}$" henceforward.

### 3.1.2    Enhanced approach

Following the suggestions by Hunziker et al. (2017), DATA$_{RAW}$ was carefully checked for systematically occurring data quality issues. An extensive set of tests (11 for TX and TN, 15 for PRCP) was applied, and flags were set for each test on an annual

time scale. Thanks to flagging each quality issue individually in the database, specific time series segments can subsequently be selected that are adequate for the intended purpose. Furthermore, for a segment of one station record (TN of Progreso (Peru), see Hunziker et al. (2017)), daily corrections were calculated, since the origin of the correctable error was unambiguously identified.

Time series segments affected by data quality issues that disturb the calculation of monthly means (temperature) and sums

(PRCP) were removed from further analyses, which reduced the number of valid measurements by about 40 %. Table 1 briefly describes the data quality issues and related thresholds that led to the exclusion of time series segments. Thresholds were chosen so that quality problems that may significantly affect the subsequent climatological analyses are excluded, whereas data containing minor problems still remain in the dataset. Note that the QC tests were applied in parallel, and therefore time series segment may be affected by several data quality issues simultaneously. If suspicious data patterns could not clearly be

attributed to a specific data quality issue, they were classified as "Irregularities in the data pattern". For details on most of the data quality issues included in the present study, see Hunziker et al. (2017).

Some data quality issues may significantly affect daily observations, but they may lose their significance by monthly aggregation. This particularly applies to observations affected by multi-day PRCP accumulations. Such data may still be adequate to calculate monthly totals (WMO, 2011) but cannot be used on a daily time scale (Viney and Bates, 2004). Therefore,

more rigorous thresholds were used for data quality issues that cause multi-day PRCP accumulations (i.e. "Small PRCP gaps" and "Weekly PRCP cycles"), if the data were later analysed on a daily time scale (Table 1). In the present study, daily data are used to analyse the correlation on a daily scale (Sect. 3.3) and the climate change indices (Sect. 3.5).

The QC tests suggested by Hunziker et al. (2017) detect data quality issues that occur systematically during longer time periods (months to years). Therefore, they are not a self-contained QC approach and should be combined with other tests. That is why the GHCN-Daily QC was additionally applied (see Sect. 3.1.1) after removing time series segments of insufficient quality for monthly aggregation. The GHCN-Daily QC added flags to approximately 0.26 % (temperature) and 0.10 % (PRCP) of the remaining observations.

This QC procedure may be considered as an enhancement of applying the GHCN-Daily QC only. Hence, the resulting dataset will be named "DATA$_{QC-E}$" henceforward.

Note that Hunziker et al. (2017) further suggest the inclusion of additional information derived from metadata into the QC process. This allows the removal of station records that were generated under inappropriate conditions, such as poor station siting or severe lack of station maintenance. The present study, however, only considers quality issues and errors that are directly detectable in the measurement data. Hence, time series of questionable quality that could be removed by including metadata in the QC process remain in the dataset.

### 3.2 Calculation of monthly and yearly means and sum

According to WMO (2011), monthly means can be calculated for continuous variables such as temperature if ≤10 daily measurements are missing. However, we used the stricter approach suggested for the calculation of monthly 30-year standard normals (WMO, 1989) that allows ≤5 missing observations (3 if in succession). For cumulative variables such as rainfall, values should be calculated only if either all daily observations are available, or if unrecorded PRCP amounts are incorporated in the next measurement (WMO, 2011). At various Bolivian stations, measurements are not taken on one day a week (usually Sundays) (Hunziker et al., 2017). This affects particularly weather stations at secondary airports that are not operating on Sundays. PRCP on these days is usually incorporated in the measurement of the next operation day. Therefore, monthly PRCP sums were calculated if ≤5 daily observations were missing and if no missing observations occurred in succession. Annual means (temperature) and sums (PRCP) were calculated based on monthly values, and yearly values were calculated only if 12 valid months were available (WMO, 2011).

For many datasets and studies, gaps in time series are filled (e.g. Auer et al., 2007; Kizza et al., 2012; Vuille et al., 2000). There are many techniques for data estimation (e.g. WMO, 2011) that may increase the time series completeness and hence data availability. However, data estimation is difficult to apply to Central Andean station records due to complex topography, sparse station networks, and mostly few observed atmospheric variables. Furthermore, the input data for ACMANT3 (homogenization method used in the present study, see Sect. 3.4.2) should best not include estimated data (Domonkos and Coll, 2017). Hence, in order to avoid introducing uncertainty by filling gaps, no data were estimated for the present study.

### 3.3 Correlation analysis

Before calculating the correlation coefficient of station pairs, time series were standardized by subtracting the mean and dividing by the standard deviation. In order to remove the influence of trends and inhomogeneities, the differences between

one observation and the next were calculated. From these time series of the first differences, Spearman rank correlations were computed for the period 1981 to 2010.

For correlations on the monthly time scale, daily observations were aggregated as described in Sect. 3.2. Only time series containing ≥80 % of valid monthly values in the 30-year long period of interest were considered. Removing the flagged observations and time series without sufficient data resulted in 98 (TX), 99 (TN), and 218 (PRCP) valid monthly station records for $DATA_{QC-S}$, and in 56 (TX), 54 (TN), and 105 (PRCP) valid monthly time series for $DATA_{QC-E}$.

To standardize measurement values on a daily time scale, daily means and standard deviations were calculated based on linear interpolation of monthly means and standard deviations. If equal values occurred in succession in the original observations, the first differences of the standardized values were set to zero in order to not bias correlation coefficients by the seasonality of the standardization.

Because unreported shifting of dates occurs frequently in the Central Andean observation networks (Hunziker et al., 2017), temporal dislocation in daily time series pairs must be considered. For example, a high correlation of two Central Andean time series of the first differences often becomes slightly negative if one of the two time series is shifted by one day. Therefore, shifts of -2 to +2 days were applied to one time series of each station pair, and the highest correlation value was expected to be the real correlation coefficient. This method may artificially increase correlations that are close to zero or negative in reality. However, such low correlations are not of interest in the present study. Furthermore, we use the median to quantify the effect of data quality issues on correlations, which eliminates the potential bias introduced to low correlations. Time series with <80 % of daily observations in the period 1981 to 2010 were removed from the daily correlation analysis. This resulted in 104 (TX), 106 (TN), and 220 (PRCP) valid daily station records available for $DATA_{QC-S}$, and in 59 (TX), 58 (TN), and 90 (PRCP) records for $DATA_{QC-E}$.

### 3.4 Data homogenization

### 3.4.1 Clustering

In order to build station groups that share a similar background climate, we applied agglomerative hierarchical clustering with complete linkage on the monthly station correlation matrices (see Sect. 3.3). Time series that did not share ≥120 common valid months with ≥10 neighbours were removed from the data homogenization process. For the break detection and adjustment method used in this study (Sect. 3.4.2), the optimal cluster size is usually around 20 to 30 stations, but the optimal number of stations can be much higher if record lengths and data completeness differ between the time series (Domonkos and Coll, 2017). This strongly applies to the Central Andean data. Therefore, we selected three clusters for TX and TN with a median size of 60 ($DATA_{QC-S}$) and 40 ($DATA_{QC-E}$) stations. For PRCP, six ($DATA_{QC-S}$) and five ($DATA_{QC-E}$) clusters were selected with a median cluster size of 65 ($DATA_{QC-S}$) and 42 ($DATA_{QC-E}$). The minimum and maximum cluster size is 11 and 94 stations, respectively.

The spatial structure of the clusters is similar for DATA$_{QC-S}$ and DATA$_{QC-E}$. For temperature, two main clusters were detected, representing the Lowlands and the Altiplano. Stations of the third cluster are located mostly along the eastern Andean slopes. Spatial illustrations of the clusters are shown in Fig. S1 in the supplementary material.

### 3.4.2 Breakpoint detection and adjustment

There are various established homogenization approaches (e.g. Aguilar et al., 2003; Ribeiro et al., 2016; Venema et al., 2012). For the present study, the method ACMANT was chosen. ACMANT is a fully automatic method that does not incorporate metadata. Hence, the approach is objective, in contrast to semi-automatic approaches such as HOMER (Mestre et al., 2013) that require various subjective decisions. This subjectivity may influence the results of the homogenization process (Vertačnik et al., 2015). For the aim of the present study to evaluate the effects of undetected data quality issues, it is important to avoid

such disturbances. ACMANT is a state of the art homogenization method having one of the best performances (Ribeiro et al., 2016; Venema et al., 2012). Recently, a new version of the approach (ACMANT3) was published (Domonkos and Coll, 2017). Compared to previous versions (Domonkos, 2011; Domonkos, 2015), the performance of the method was further improved and the range of use increased (Domonkos and Coll, 2017).

ACMANT3 includes a recommended function for detecting monthly outliers that was applied before detecting and correcting

breakpoints. About twice as many monthly outliers were detected in DATA$_{QC-S}$ than in DATA$_{QC-E}$. The highest frequency of monthly outliers was found in TN of DATA$_{QC-S}$ with 0.16 outliers per decade. All monthly outliers were removed from DATA$_{QC-S}$ and DATA$_{QC-E.}$

### 3.5 Trend calculation

Trends of annual values and climate change indices were analysed for the entire study area in the 30-year time period 1981 to

2010. However, trend signals differ between the varied climate zones covered by the DECADE dataset. Therefore, we decided to focus particularly on the Altiplano region for trend analyses. Time series from the Altiplano that satisfy the completeness requirements originate nearly exclusively from stations located in the north-western Bolivian department of La Paz and the adjacent Peruvian department of Puno. In this spatially limited region, the station network is dense compared to the rest of the study area (Fig. 1). Therefore, relatively homogeneous trend signals may be expected.

Magnitudes of linear trends were calculated with the Theil-Sen estimator, which is calculated by the median of the slopes of all data pairs of a time series (Sen, 1968; Theil, 1950). The method is more insensitive to outliers and more robust than other trend estimators such as Ordinary Least Squares. For individual station records, the significance of trends is not of major interest in the present study and was therefore not tested. Furthermore, taking serial correlation into account in trend tests would cause large uncertainties due to the missing values in the time series. However, for the Altiplano stations, trends of

spatially averaged anomalies were tested with the Mann-Kendall test at the 5 % significance level. Before applying the Mann-Kendall test, the time series were pre-whitened (Frei, 2013) in order to remove the influence of serial correlation.

Trends of annual means (temperature) and sums (PRCP) were analysed based on yearly aggregated data (see Sect. 3.2). Time series with <80 % valid yearly values 1981 to 2010 were removed previously. This resulted in 54 (TX), 48 (TN), and 105 (PRCP) valid annual station records for DATA$_{QC-S}$, and in 40 (TX), 29 (TN), and 48 (PRCP) annual time series for DATA$_{QC-E}$. In order to investigate the effect of undetected data quality issues on extremes, we computed the frequently used climate change

indices defined by the CCl/CLIVAR/JCOMM Expert Team on Climate Change Detection and Indices (ETCCDI) (http://etccdi.pacificclimate.org/list_27_indices.shtml) for 1981 to 2010. For the calculation of the indices, we used the software tool RClimDex (Zhang and Yang, 2004) that is often applied in climatological studies (e.g. Kioutsioukis et al., 2010; Kruger and Sekele, 2013; New et al., 2006). RClimDex calculates monthly (yearly) index values if ≤3 (≤15) observations are missing (Zhang and Yang, 2004). The indices discussed in the present study are namely the diurnal temperature range (DTR),

cool days (TX10p), cool nights (TN10p), warm days (TX90p), warm nights (TN90p), frost days (FD), annual contribution from very wet days (R95pTOT), and the simple daily intensity index (SDII) (Table 2). Note that all indices were calculated on an annual scale. For indices based on percentiles, the baseline period was calculated from the 30-year period 1981 to 2010. Indices units in percentage were converted to days per year.

The ETCCDI climate change indices describe moderate to very moderate extreme events that occur usually many times per

15   year. Therefore, they are particularly suitable for the application on short time series. For the index calculation of the homogenized datasets, daily measurements were corrected by adding monthly adjustment values (temperature) and by multiplying with monthly adjustment factors (PRCP) that were computed with ACMANT3. Applying monthly corrections on a time series does not guarantee homogeneity on a daily time scale (Brönnimann, 2015; Costa and Soares, 2009; Trewin, 2013). However, since the present study aims to compare the effects of different QC methods, potential deficits in adjusting

daily observations with monthly factors do not bias the results. Considering the large and frequent inhomogeneities detected in the Central Andean time series (Sect. 4.3), the homogeneity of the ETCCDI climate change indices will most likely be increased strongly by correcting the daily time series with the monthly adjustment values.

Trends of the ETCCDI climate change indices were only calculated for time series with ≥80 % of valid yearly index values in the period 1981 to 2010. For the analyses of the climate change indices, about 50 (DATA$_{QC-S}$) and 30 (DATA$_{QC-E}$) valid time

series for the temperature derived indices (TX10P, TX90P, TN10P, TN90P, and FD) were available. For DTR that depends on both TX and TN observations, 41 (DATA$_{QC-S}$) and 22 (DATA$_{QC-E}$) time series could be analysed. For the PRCP derived indices SDII and R95pTOT, 106 (DATA$_{QC-S}$) and 38 (DATA$_{QC-E}$) indices time series were available.

## 4   Results

### 4.1   Frequency of data quality issues

The frequency of systematic data quality issues clearly varies between the different regions (Table 1). Overall, data quality issues occur least frequently in the Lowlands. Many weather stations in this area are located at airports and are operated by the Bolivian civil airport administration (Hunziker et al., 2017). The personnel at the airports are generally better trained in taking

observations than the laypersons running most of the other weather stations in the Central Andean area. In contrast, data quality issues occur most frequently in the Valleys. Many of these stations are located in rather remote regions, and they generally get less attention by the network operators than other stations in the network.

Some systematic data quality issues are relevant in one region, but not in another. For instance, the "missing temperature intervals" are important in TN observations in the Altiplano and the Valleys, but do barely occur in the Lowlands. This problem usually occurs in measurements around 0 °C. Temperatures in the Lowlands rarely drop to the freezing point, and hence this issue is largely absent. In contrary, "Weekly PRCP cycles" occur particularly often in the Lowlands, where the fraction of observations at airports is large (secondary airports are usually out of service on Sundays).

The data quality issue "Irregularities in the data pattern" reaches the threshold for exclusion of time series segments more often than the other quality problems. This error classification combines all suspicious data patterns that cannot be clearly classified as another quality issue. In contrast to other data quality issues, irregularities in the data pattern occur in all regions. Time series segments of rather low quality are often affected by several problems simultaneously, which includes usually rather unspecific irregularities in the data pattern.

Overall, the quality of the TX, TN, and particularly PRCP observations slightly increased in the last decades (Fig. 2). However, the frequency of some data quality issues increased, such as strong "Asymmetric rounding patterns" in TX and TN observations, or "Missing temperature intervals" in TN time series. There is no strong or abrupt change of the frequency of the data quality issues between 1981 and 2010. The same applies to the temporal development of data quality issues in the single regions Altiplano, Valleys, and Lowlands (not shown).

## 4.2 Correlation analysis

Detecting and removing erroneous measurement values and time series segments affects the correlation of station pairs in two ways. On one hand, time series may not anymore fulfil the completeness requirements in the time period of interest. This occurs more often when applying the enhanced than the standard QC approach. While highly correlated station records remain in DATA$_{QC-E}$, the enhanced QC largely removes the low correlation coefficients found in DATA$_{QC-S}$ (Fig. 3). Hence, data quality issues that are undetected by the standard QC method result in low correlation coefficients of station pairs. On the other hand, the correlations of station pairs may change if rather short time series segments are removed due to data quality problems. Usually, this results in an increase of the correlation coefficients (Fig. 3), which may reach up to 0.07 (TX) and 0.09 (PRCP) on a monthly and daily time scale. For TN, maximum correlation improvements are 0.10 and 0.05 on a monthly and a daily time scale, respectively. Since this study only includes time series with ≥80 % of valid values, each time series pair shares ≥60 % of common observations between 1981 and 2010 (i.e. ≥18 years).

The resulting median differences of correlation coefficients in DATA$_{QC-S}$ and DATA$_{QC-E}$ are relatively constant up to station distances of approximately 300 km (Fig. 4, Fig. 5). The overall differences between DATA$_{QC-E}$ and DATA$_{QC-S}$ are 0.15 (TX), 0.24 (TN), and 0.11 (PRCP) on a monthly, and 0.10 (TX) and 0.13 (TN) on a daily time scale (Table 3). For daily PRCP, median correlation coefficients converge quickly to zero with increasing station distance, and therefore stations within a

100 km radius were analysed. The resulting median correlation difference for daily PRCP between $DATA_{QC-E}$ and $DATA_{QC-S}$ is 0.06.

However, the effect of undetected data quality issues on station correlations varies strongly between the different regions. While it is small in the Lowlands, it is very pronounced in the Valleys. This can be partly explained by the high fraction of station records affected by severe data quality issues in the Valleys. Lowland stations, in contrast, are often located at airports where data quality problems occur less frequently.

There are remarkable differences between median correlation coefficients of station pairs in the Lowlands, the Valleys, and the Altiplano (Fig. 4, Fig. 5, Table 3). This is primarily explicable by the varied topography. While the Lowlands are largely flat, the topography of the Altiplano and the Valleys is moderately and highly complex, respectively. Therefore, the median correlations are overall highest in the Lowlands, and lowest in the Valleys.

However, spatial correlations are further modulated by regional weather and climate characteristics. For instance, PRCP correlation coefficients in the Altiplano are higher than in the Lowlands on a monthly time scale, whereas the opposite applies to correlations on a daily time scale (Fig. 4, Fig. 5, Table 3). On one hand, the Altiplano receives precipitation from deep convective storms during austral summer (Garreaud, 2009), and wet periods tend to cluster in episodes of about a week, interrupted by dry spells of similar duration (Garreaud, 1999). On the other hand, cold surges in the Lowlands occur with a periodicity of approximately one to two weeks (Garreaud, 2000) and usually last two or three days (Espinoza et al., 2013). In summertime, these events cause synoptic-scale bands of enhanced and suppressed deep convection that structure temporal PRCP occurrence (Garreaud, 2000). Hence, rain events in the Altiplano cluster on clearly larger time scales than in the Lowlands. This favours high correlation of monthly PRCP sums in the Altiplano and high correlation coefficients of daily observations in the Lowlands. Note, however, that the correlation differences between the regions are more pronounced within $DATA_{QC-S}$ than within $DATA_{QC-E}$.

## 4.3 Data homogenization

One out of three TN station clusters of $DATA_{QC-S}$ (34 station records) could not be homogenized because of too low spatial-temporal coherence. Most of the time series in this cluster are affected by systematic data quality issues that were not detected with the standard QC approach. Since these station records could not be homogenized, they were excluded from all further analyses.

ACMANT3 detected a high number of breakpoints in the station records. For temperature, about one breakpoint per decade was detected on average, with a slightly higher breakpoint frequency in $DATA_{QC-S}$ than for $DATA_{QC-E}$ (Table 4). For PRCP, 0.3 breakpoints per decade were found in $DATA_{QC-S}$ and 0.2 in $DATA_{QC-E}$. Median, mean and maximum breakpoint sizes are clearly larger in $DATA_{QC-S}$ than $DATA_{QC-E}$ for all climate variables (Table 4).

Adjustments values (temperature) and factors (PRCP) close to zero (temperature) and one (PRCP) are more frequent for $DATA_{QC-E}$ than for $DATA_{QC-S}$ (Fig. 6). Furthermore, maxima of the absolute adjustments are higher for $DATA_{QC-S}$ than $DATA_{QC-E}$, reaching up to 10.2 °C (temperature) and 3.5 (PRCP). However, there are not only differences in the standard

deviation, but also in the symmetry of the adjustment distributions. For example, the adjustment factors for PRCP of $DATA_{QC-E}$ indicate a density peak at around 1.3, which is not found for $DATA_{QC-S}$ (Fig. 6). For TN of the Altiplano stations in 1981 to 2010, there is a high density of adjustments values around -1 °C. This peak is more pronounced in $DATA_{QC-E}$ than in $DATA_{QC-S}$. As a result, the median adjustment in $DATA_{QC-E}$ is -0.5 °C, whereas it is +0.2 °C in $DATA_{QC-S}$. The same median
adjustments are calculated for complete record lengths of the Altiplano stations (not shown), indicating the detection of an overall warm bias in earlier TN observations of $DATA_{QC-E}$ but not of $DATA_{QC-S}$.

Henceforward, the homogenized datasets $DATA_{QC-S}$ and $DATA_{QC-E}$ are named "$DATA_{QC-S\_H}$" and "$DATA_{QC-E\_H}$", respectively. Note that some time series segments could not be homogenized due to lacking references stations with the required correlation. For the trend analyses, all time series segments that remained unhomogenized were also excluded from
the unhomogenized datasets (i.e. $DATA_{QC-S}$ and $DATA_{QC-E}$) in order to maintain the comparability between unhomogenized and homogenized datasets.

### 4.4 Trends

### 4.4.1 Annual temperature averages

Overall, there is a clear positive TX trend in the entire study area (Fig. 7). The few negative TX trends in the unhomogenized
station records disappear due to data homogenization. In the Altiplano, the trend of the spatially averaged anomalies is significant and varies between +0.40 ($DATA_{QC-E\_H}$) and +0.44 °C ($DATA_{QC-E}$) per decade (Table 5). TN trends, however, are more ambiguous. Spatial trend patterns are unclear, except for $DATA_{QC-E\_H}$, where a clear warming is found in the north-eastern Altiplano, and slight cooling in the south and the Lowlands. This pattern is spatially coherent and substantially diverges from the spatial trend patterns derived from the other datasets. As a result, TN trends of spatially averaged anomalies calculated
from $DATA_{QC-E\_H}$ in the Altiplano are significant with +0.22 °C per decade, whereas they are close to zero and insignificant if calculated from the other datasets (Table 5). This may be at least partly ascribed to the results of the data homogenization process, which suggest a clear overall warm bias in earlier TN observations in the Altiplano in $DATA_{QC-E}$, but not in $DATA_{QC-S}$ (Sect. 4.3).

The spread of individual station trends is slightly lower in $DATA_{QC-E}$ than in $DATA_{QC-S}$ (Fig. 8). However, the spread of trends
is much more reduced by data homogenization than by enhancing the QC approach. For TX, the trend spreads derived from the homogenized datasets $DATA_{QC-S\_H}$ and $DATA_{QC-E\_H}$ are similar, whereas they strongly differ for TN. The TN trend spread of the entire study area derived from $DATA_{QC-S\_H}$ is small and ranges between +0.02 and +0.09 °C per decade within the 25[th] and 75[th] percentile. In contrast, data homogenization of $DATA_{QC-E}$ does not cause such a pronounced decrease of the trend spread.

### 4.4.2 Annual precipitation sums

PRCP trends are negative for most station records (Fig. 7). The spatial pattern of trend magnitudes is more coherent if trends are calculated from $DATA_{QC-E\_H}$ than from the other datasets. Despite the previous homogenization of the time series in $DATA_{QC-S\_H}$, there are strong positive and negative trends of stations within short distance. For all regions (Lowlands, Valleys, and Altiplano), trends of the spatially averaged anomalies are negative, particularly if derived from $DATA_{QC-E\_H}$ (not shown). However, these trends are barely significant due to the high interannual variability of PRCP.

The trend spread and frequency of very strong trends is lower in $DATA_{QC-E}$ than in $DATA_{QC-S}$ (Fig. 8). Data homogenization reduces the trend spread of the PRCP time series, but considerably less than for temperature data. Overall, the spread of PRCP trends of individual station records is relatively large in all datasets.

### 4.4.3 Climate change indices

Trends of the median diurnal temperature range (DTR) of all datasets are positive (Fig. 9). The spread of trends calculated from the unhomogenized datasets is large, ranging from -1.21 to +2.17 °C per decade. It is lower for $DATA_{QC-E}$ than for $DATA_{QC-S}$, particularly on regional scale such as in the Altiplano (Fig. 10). However, data homogenization is most relevant for increasing the coherency of DTR trends. This is particularly remarkable for $DATA_{QC-E\_H}$ in the Altiplano, where individual station trends of the DTR are reduced to a range between 0.10 and 0.29 °C per decade (Fig. 10). Besides this high DTR trend coherency derived from $DATA_{QC-E\_H}$ in the Altiplano, trend magnitudes are clearly lower than those derived from the other datasets. This manifests in an insignificant trend of the spatially averaged anomalies of +0.23 °C per decade for $DATA_{QC-E\_H}$, whereas the trends calculated from the other datasets are all significant and range between +0.39 ($DATA_{QC-S\_H}$) and +0.56 °C per decade ($DATA_{QC-E\_H}$).

The overall trend signal of the TX based percentile indices TX10p and TX90p is relatively uniform among the different datasets, indicating a reduction of cool days and an increase of warm days (Fig. 9, Fig. 10). The trends of the spatially averaged anomalies in the Altiplano calculated from the different datasets are all significant and range between -11.9 ($DATA_{QC-E\_H}$) and -14.4 ($DATA_{QC-S\_H}$) cool days per decade, and between +8.7 ($DATA_{QC-S}$) and +11.0 ($DATA_{QC-S\_H}$) warm days per decade (Table 5). For both indices, the trend magnitudes are more pronounced for $DATA_{QC-S\_H}$ than for $DATA_{QC-E\_H}$.

Median trend magnitudes of TN based percentile indices (TN10p, TN90p) are smaller than those based on TX, and the spreads of individual station trends are larger, particularly in the Altiplano (Fig. 9, Fig. 10). In this region, trend magnitudes derived from $DATA_{QC-E\_H}$ differ substantially from the other datasets (Fig. 10) by indicating a clear warming trend in all indices (i.e. decrease of cool nights and frost days, increase of warm nights). This is confirmed by the trends of the spatially averaged anomalies that indicate a significant decrease of cool nights (-5.8 days per decade) and a significant increase of warm nights (+8.8 days per decade) (Table 5). In contrast, the trends calculated from the other datasets are all insignificant and have lower trend magnitudes. The same pattern is found for trends of the frequency of frost days (FD). Trends calculated from $DATA_{QC-E\_H}$

indicate a significant decrease of FD (-6.5 days per decade), whereas the trends of the other datasets are insignificant and close to zero (Table 5).

The PRCP based climate change indices indicate a slight decrease in the annual contribution of very wet days (R95pTOT) and a decreasing intensity of precipitation events (SDII), particularly in the Altiplano. This signal is more pronounced for the

datasets quality controlled with the standard method than for the datasets quality controlled with the enhanced approach. Trends of the spatially averaged anomalies are not significant, except for the trends of SDII derived from the dataset quality controlled with the standard approach in the Altiplano (i.e. -3.1 mm day$^{-1}$ (DATA$_{QC-S}$) and -2.1 mm day$^{-1}$ (DATA$_{QC S\_H}$) per decade). In contrast to the indices derived from temperature data, applying the enhanced QC approach reduces the trend spread of the PRCP based indices more than statistical data homogenization. Compared to DATA$_{QC-S}$, the standard deviation of the relative

trends in the complete study area calculated from DATA$_{QC-S\_H}$, DATA$_{QC-E}$, and DATA$_{QC-E\_H}$ are 20, 40, and 50 % lower, respectively.

## 5 Discussion

Systematically occurring data quality issues affect a large fraction of Central Andean station records, making about 40 % inadequate for the calculation of monthly means (temperature) and sums (PRCP). The frequency of such problems may vary

strongly in space and time. Systematic data quality issues remain largely undetected when applying standard data quality control methods such as the one used for GHCN-Daily. Hence, important data sources may be affected substantially by such undetected data quality issues (abbreviated as UDQI henceforward). Including tests to specifically detect such erroneous patterns could significantly increase the quality of many datasets.

On a monthly time scale and up to 300 km station distance, UDQI cause a reduction of median correlation coefficients by 0.15

(TX), 0.24 (TN), and 0.11 (PRCP) compared to unaffected data. On a daily time scale, this reduction is 0.10 (TX) and 0.13 (TN) for station pairs within 300 km distance, and 0.06 (PRCP) for stations within 100 km distance. These findings confirm the assumption by Gubler et al. (2017) that the strong differences in correlation coefficients between station networks of the Peruvian Andes and Switzerland may not be explained by unequal climate regimes alone. Hypothesising that UDQI occur more frequently in station networks of developing than developed countries, a higher frequency of such errors can be expected

in tropical areas than in mid-latitudes. Making this assumption, UDQI may partly explain the particularly low correlation decay distances in the tropics described by New et al. (2000). Using relatively high minimum correlation thresholds in climatological analyses (e.g. data homogenization) may reduce the amount of station records affected by UDQI. As a more advanced approach, weighting correlation coefficients with station distances (i.e. more weight to station pairs further away from each other for equal correlation coefficients) could particularly take UDQI into account.

UDQI induce additional inhomogeneities in observational record. The resulting decrease in the signal-to-noise ratio may decrease the performance of statistical data homogenization methods (Domonkos, 2013). This is particularly problematic in sparse observational networks, where a high number of breakpoints may result in adjustments that deteriorate the temporal

consistency of station records (Gubler et al., 2017). In the Central Andean region, UDQI increase the number of statistically detected breakpoints by about 15 % for TX and TN, and by 50 % for PRCP. They also increase the median breakpoint size by 35 to 40 % (TX and TN) and 60 % (RPCP), and increase break size maxima by up to 100 % (temperature) and 70 % (PRCP). Since UDQI have larger relative effects on break sizes than on the number of detected breakpoints, they apparently deteriorate

the detectability of small non-climatic inhomogeneities.

The effect of UDQI also manifests in the adjustment values (temperature) and factors (PRCP) resulting from the data homogenization process. UDQI cause a reduction in the frequency of small adjustments and an increase of large adjustments. They also may induce an adjustment bias. For instance, the median adjustment value for TN station records in the Altiplano is -0.5 °C. If the same dataset contains UDQI, the resulting median adjustment is +0.2 °C. This difference of the adjustment

could be caused in two ways. First, UDQI may introduce a systematic bias (a cold bias in earlier observations in this case). This would require the occurrence of certain types of UDQI in many station records of a dataset which would cause a systematic bias and which would strongly change their frequency in time. For the Central Andean area, however, there is no clear indication that UDQI meet these requirements in the period 1981 to 2010. Second, UDQI may not introduce a bias by themselves, but they impede the detection of an existing bias (warm bias in earlier observation in case of TN in the Altiplano)

by introducing artificial noise. Such a warm bias could have been introduced, for example, by location changes of weather stations to systematically different sites (e.g. further away from buildings). The second possibility seems to be the more likely cause of the observed adjustment differences of TN records in the Altiplano. Hence, UDQI may impede the adjustments of systematic biases introduced by inhomogeneities. In summary, UDQI may substantially decrease the performance of statistical data homogenization methods.

Between 1981 and 2010, a pronounced and relatively uniform increase of global mean temperatures was observed (IPCC, 2013). Similarly, clear overall warming trends in the same period were reported from analyses of extremes indices (Donat et al., 2013b). Hence, 1981 to 2010 is a suitable period for analysing linear temperature trends, and clear trend signals may be expected in the Central Andean area too.

UDQI increase the overall spread of individual station trends. Statistical data homogenization may largely reduce or eliminate

this effect, but only at the cost of more and larger breakpoints, which lowers the performance of data homogenization methods. For instance, the trend spread of homogenized TN time series quality controlled with the standard approach ($DATA_{QC-S\_H}$) is extremely small (Fig. 8). This clearly deviates from the trend spreads observed for TX, as well as from the trend spread derived from $DATA_{QC-E\_H}$. Hence, station trends computed from $DATA_{QC-S\_H}$ seem rather implausible and may indicate an over-homogenization. In contrast, the low trend spread of diurnal temperature range (DTR) derived from $DATA_{QC-E\_H}$ (Fig. 10)

suggests that the independent data homogenization of TX and TN observations are consistent with each other. Furthermore, trends calculated from $DATA_{QC-E\_H}$ are spatially more coherent than those derived from $DATA_{QC-S\_H}$, particularly for TN and PRCP (Fig. 7).

If datasets contain UDQI and/or are unhomogenized, TN trends of averaged anomalies in the Altiplano are close to zero and insignificant, and trends of the diurnal temperature range (DTR) are strongly positive and significant at the 5 % level (Table 5).

On the contrary, TN trends derived from $DATA_{QC-E\_H}$ are significantly positive, and trends of the DTR are insignificant. Hence, mean temperature trends in the Altiplano are more in accordance with the global observations (IPCC, 2013) if systematically occurring data quality issues are removed from the dataset. Nevertheless, the influence of UDQI in station records from the Altiplano explains roughly half of the trend difference between TX and TN. Hence, there must be other factors (climatological

or non-climatological) that cause a stronger increase in TX than TN in the Altiplano. An indication for a climatological explanation of the positive DTR trends are the simultaneously observed negative PRCP trends. Several authors have described a negative correlation between DTR and PRCP trends (Dittus et al., 2014; Jaswal et al., 2016; Zhou et al., 2009). Hence, the observations in the Altiplano would be in accordance with these findings.

Since systematically occurring data quality issues especially affect extremes (Hunziker et al., 2017), UDQI have a stronger

effect on ETCCDI climate change indices than on trends of annual means (temperature) and sums (PRCP). Particularly in the Altiplano, the spread of individual station trends is usually more coherent if trends are calculated from $DATA_{QC-E\_H}$ than from the other datasets. For the climate change indices derived from PRCP analysed in the present study (R95pTOT and SDII), most very strong trends of individual time series are caused by UDQI and cannot be adjusted by statistical data homogenization. Hence, highly incoherent spatial trend signals and strong differences in trend magnitudes between neighbouring stations as

detected in many studies (e.g. Skansi et al., 2013; Vuille et al., 2003; You et al., 2011) may potentially be ascribed to UDQI. The overall temperature trend signals derived from $DATA_{QC-E\_H}$ in the Altiplano are highly coherent, indicating significant warming throughout all indices. This trend pattern of moderate extreme events is more in accordance with the global observations (e.g. IPCC, 2013) than the trend patterns derived from the other datasets. Consequently, UDQI may at least partly explain the discrepancies of trends detected in the Altiplano compared to most other world regions.

According to Donat et al. (2013b), gridding observations minimizes the impact of data quality issues at individual stations due to averaging. This, however, may not be true if UDQI cause systematic biases. We calculated trends of the relevant climate change indices derived from the two 2.5° x 3.75° grid cells of the HadEX2 dataset (Donat et al., 2013b) that are most representative for the Altiplano. Overall, these trends in the period 1981 to 2010 are most similar to the trends derived from $DATA_{QC-S\_H}$. However, the trends of a few climate change indices derived from HadEx2 have extreme magnitudes, such as a

detected increase of +14.0 frost days per decade in one of the grid cells. Furthermore, virtually all of these trends are insignificant due to large variabilities in the annual index time series. Even though these findings do not allow drawing of clear conclusions, they suggest that UDQI affect the dataset and influence the trend calculations.

The quantifications of the effects of UDQI presented in this study are rather an estimate of the lower limit, since the enhanced QC method applied here may still not have detected and removed all relevant data quality issues. Furthermore, metadata were

not accessed as information source for QC, which may help to detect and remove additional time series segments of inadequate quality (Hunziker et al., 2017). The quantifications presented in this article cannot be generalized to global datasets. The frequencies and characteristics of data quality issues occurring in manned station networks depend on various factors, such as the observing practices, the capabilities of the personnel, or the data transcription procedures. Furthermore, the wide range of QC approaches applied in National Weather Services will detect different fractions of errors and data quality issues. The effects

of UDQI on climatological analyses also depend on the climate regime. For instance, missed measurements of small precipitation events of up to a few millimetres may only have a negligible effect on monthly sums in wet regions (e.g. Amazonian Lowlands), whereas they may significantly bias monthly PRCP sums in rather dry areas (e.g. Altiplano) due to low overall PRCP and evaporation losses. As demonstrated in this work, UDQI have stronger effects on climatological

analyses derived from TN than TX observations. On one hand, TN is generally more variable and spatially heterogeneous than TX (Luhunga et al., 2014; Mahmood et al., 2006; New et al., 1999). On the other hand, measurement errors may occur more frequently in TN than TX. For example, TN falls naturally more often below freezing temperature than TX, and temperature values around and below 0 °C often trigger measurement errors by observers as well as data transcription errors (Hunziker et al., 2017). Hence, frequency and effects of UDQI also vary spatially and temporally.

Removing a relatively large fraction of observations (such as 40 % in the present study) from a dataset may affect the results of climatological analyses. Reducing the spatial density of available data normally decreases the quality of the results such as for data homogenization (Caussinus and Mestre, 2004; Domonkos, 2013; Gubler et al., 2017). With the present study, however, we have demonstrated that removing time series segments affected by UDQI increase the overall quality of the dataset, and results of climatological analyses are consequently more coherent and reliable. The disadvantage of fewer available

observation is outperformed by the quality increase of the dataset.

## 6 Conclusions

Systematically occurring data quality issues may affect large fractions of time series in observational datasets. In the Central Andean area, about 40 % of the observations are inappropriate for the calculation of monthly temperature means and precipitation sums. These problems remain largely undetected by standard quality control methods. In the present study, we

applied a standard and an enhanced quality control approach on the same dataset. The enhanced approach should particularly detect systematically occurring data quality issues. We subsequently compared the results of various climatological analyses derived from data quality controlled with the two different methods.

Undetected data quality issues (UDQI) substantially lower the correlation coefficients of station pairs. This directly affects various methods such as clustering or data homogenization.

The performance of data homogenization approaches deteriorates if time series contain UDQI. Since UDQI induce inhomogeneities in time series, they increase the number and average size of breakpoints in the data. As a result of the increased noise in the station records, the skill of statistical data homogenization methods to detect and correct smaller inhomogeneities is reduced. Furthermore, data homogenization approaches may fail to detect and correct systematic biases caused by inhomogeneities due to UDQI. In the Altiplano for instance, a median adjustment value of -0.5 °C for minimum temperature

observations was detected for time series free of UDQI, whereas a median adjustment of +0.2° C was computed for the station records affected by UDQI. This warm bias in earlier TN observations may affect previous studies using station records from the Altiplano. Hence, data homogenization methods rely on data that are largely free of UDQI in order to perform satisfactorily.

Removing UDQI from a dataset increases the spatial coherence and reduces the spread of individual stations trends. Furthermore, UDQI may systematically bias trends. For instance, regional minimum temperature trends in the Altiplano are insignificant and close to zero if calculated from station records affected by UDQI, whereas trends are significant and clearly positive if derived from time series free of UDQI.

Since UDQI especially affect extremes, they are particularly problematic for analysing trends of rare events such as for the ETCCDI climate change indices. In the Altiplano, trends of various indices based on minimum temperature differ significantly if derived from a dataset affected or unaffected by UDQI. For some climate change indices based on precipitation, extreme trend magnitudes at individual stations can be corrected by previously removing UDQI from the dataset, but not by statistical data homogenization.

Most likely, the results of various studies are affected by UDQI. If quality control approaches are enhanced and UDQI removed, results of climatological analyses may become more coherent and reliable. Note that neither an enhanced and comprehensive quality control can substitute for appropriate data homogenization nor vice versa.

**Acknowledgements**

This work is part of the project 'Data on climate and Extreme weather for the Central AnDEs' (DECADE), no.
IZ01Z0_147320, which is financed by the Swiss Program for Research on Global Issues for Development (r4d). It was also supported by the EU Horizon 2020 EUSTACE project (Grant Agreement 640171). We thank Peter Domonkos for the support on ACMANT3, and Xuebin Zhang and Yang Feng for providing the newest version of RClimDex. We also thank the two anonymous reviewers for their helpful comments and suggestions.

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

**Table 1: Description of systematic data quality issues and their frequencies in the DECADE database (station records with >20 years of observations) between 1981 and 2010. If not specified, the frequencies of data quality issues apply to daily observations and monthly aggregations. Frequencies of the data quality issues in maximum (TX) and minimum (TN) temperature, and precipitation (PRCP) observations are shown for the different regions (Altiplano, Valleys, and Lowlands, see Fig. 1). Thresholds leading to the exclusion of data were chosen so that data quality issues should not affect the subsequent climatological analyses of the daily and monthly aggregated data. For other analyses, these thresholds may not be adequate and consequently the frequencies of data quality issues may differ. Tests were done in parallel, and time series segments may therefore be affected by several data quality issues simultaneously. For a detailed description of frequent data quality issues, see Hunziker et al. (2017).**

| Data quality issue | Description | Threshold leading to exclusion | Frequency [%] | | |
|---|---|---|---|---|---|
| | | | Altiplano | Valleys | Lowlands |
| Missing temperature intervals | Observations within a temperature interval are missing or occur with a clearly reduced frequency | Interval of missing temperature observations >1 °C | TX: 1.0 TN: 7.4 | TX: 0.3 TN: 6.9 | TX: 0.3 TN: 0.0 |
| Rounding errors | Rounding errors in the conversion from degrees Fahrenheit to degrees Celsius (may also indicate further errors in the data) | Any error in the rounding | TX: 0.0 TN: 0.0 | TX: 4.3 TN: 2.4 | TX: 1.9 TN: 1.9 |
| Asymmetric rounding patterns | Numbers in the decimal places are not equally distributed and occur in an asymmetric form | Asymmetry in rounding pattern is strong | TX: 8.8 TN: 8.2 PRCP: 10.0 | TX: 9.6 TN: 11.0 PRCP: 13.7 | TX: 5.1 TN: 4.2 PRCP: 6.7 |
| Low measurement resolution | The reported resolution of the measurements is low | Reported measurement resolution >1 °C and >1 mm | TX: 0.0 TN: 0.0 PRCP: 0.1 | TX: 4.1 TN: 3.4 PRCP: 0.0 | TX: 1.4 TN: 1.4 PRCP: 0.0 |
| Irregularities in the data pattern | Obviously erroneous patterns in the data that cannot be classified as another data quality issue (e.g. all values in a very narrow range, randomly and strongly varying variance, truncation of negative temperatures) | Irregularities in the data pattern are moderate or strong | TX: 31.6 TN: 28.5 PRCP: 37.7 | TX: 42.3 TN: 42.2 PRCP: 42.9 | TX: 10.9 TN: 15.8 PRCP: 21.4 |
| Obvious in-homogeneities | Inhomogeneities that are large enough to be clearly identified visually as non-climatic and that occur frequently within a time series segment (i.e. inhomogeneities that are hardly correctable with data homogenization methods) | Inhomogeneities are large and occur frequently | TX: 7.7 TN: 2.5 PRCP: 2.3 | TX: 13.5 TN: 12.6 PRCP: 1.9 | TX: 0.8 TN: 0.8 PRCP: 3.2 |
| Heavy PRCP truncations | Observations of heavy PRCP events are truncated or their frequency is clearly reduced above a certain threshold | Heavy PRCP events are partially or completely truncated | PRCP: 13.3 | PRCP: 12.5 | PRCP: 5.2 |
| Small PRCP gaps | Small PRCP events are not reported, leading to a gap or a frequency reduction in values below a certain threshold | Partial and complete small PRCP gaps >5 mm (monthly) and >2 mm (daily) | PRCP: 3.0 (monthly) PRCP: 15.2 (daily) | PRCP: 7.5 (monthly) PRCP: 29.9 (daily) | PRCP: 9.5 (monthly) PRCP: 21.9 (daily) |
| Weekly PRCP cycles | The occurrence of PRCP events (wet days) significantly differs between the days of the week | Weekly PRCP cycles are strong (relaxation for monthly aggregated data, if cycle pattern indicates regularly missed observations followed by accumulation the next day) | PRCP: 0.0 (monthly) PRCP: 1.3 (daily) | PRCP: 1.8 (monthly) PRCP: 2.1 (daily) | PRCP: 2.6 (monthly) PRCP: 3.2 (daily) |

**Table 2: ETCCDI climate change indices analysed in the present study. Note that all indices were calculated on an annual time scale. Index units in percentage were converted to days per year in the following analyses.**

| ID | Index name | Index definition | Unit |
|---|---|---|---|
| DTR | Daily temperature range | Monthly mean difference between TX and TN | days |
| TX10p | Cool days | Percentage of days when TX < 10$^{th}$ percentile | % |
| TN10p | Cool nights | Percentage of days when TN < 10$^{th}$ percentile | % |
| TX90p | Warm days | Percentage of days when TX > 90$^{th}$ percentile | % |
| TN90p | Warm nights | Percentage of days when TN > 90$^{th}$ percentile | % |
| FD | Frost days | Annual count of days when TN <0 °C | days |
| R95pTOT | Annual contribution from very wet days | Annual total of daily PRCP when PRCP >95$^{th}$ percentile | mm |
| SDII | Simple precipitation intensity index | PRCP sum on wet days (PRCP ≥1 mm) divided by the number of wet days | mm day$^{-1}$ |

**Table 3: Monthly and daily median correlation coefficients of station pairs within a 300 km radius (100 km for daily PRCP) for maximum temperature (TX), minimum temperature (TN), and precipitation (PRCP).**

| | | | TX | | TN | | PRCP | |
|---|---|---|---|---|---|---|---|---|
| | | | $DATA_{QC-S}$ | $DATA_{QC-E}$ | $DATA_{QC-S}$ | $DATA_{QC-E}$ | $DATA_{QC-S}$ | $DATA_{QC-E}$ |
| Monthly | | all stations ≤300 km | 0.53 | 0.68 | 0.39 | 0.63 | 0.34 | 0.45 |
| | | Altiplano stations ≤300 km | 0.68 | 0.72 | 0.57 | 0.64 | 0.45 | 0.50 |
| | | Valley stations ≤300 km | 0.46 | 0.60 | 0.34 | 0.61 | 0.31 | 0.35 |
| | | Lowland stations ≤300 km | 0.76 | 0.79 | 0.72 | 0.75 | 0.33 | 0.36 |
| Daily | | all stations ≤300 km for temperature ≤100 km for PRCP | 0.25 | 0.35 | 0.14 | 0.27 | 0.13 | 0.19 |
| | | Altiplano stations ≤300 km for temperature ≤100 km for PRCP | 0.26 | 0.31 | 0.24 | 0.28 | 0.14 | 0.18 |
| | | Valley stations ≤300 km for temperature ≤100 km for PRCP | 0.26 | 0.45 | 0.12 | 0.22 | 0.12 | 0.24 |
| | | Lowland stations ≤300 km for temperature ≤100 km for PRCP | 0.55 | 0.59 | 0.40 | 0.44 | 0.28 | 0.32 |

5    **Table 4: Breakpoint frequencies and break sizes. For minimum and maximum temperature (TX and TN, respectively), absolute break size values in °C are shown. For precipitation (PRCP), the factors of the break sizes are indicated.**

| | TX | | TN | | PRCP | |
|---|---|---|---|---|---|---|
| | $DATA_{QC-S}$ | $DATA_{QC-E}$ | $DATA_{QC-S}$ | $DATA_{QC-E}$ | $DATA_{QC-S}$ | $DATA_{QC-E}$ |
| Breakpoints per decade | 1.0 | 0.9 | 1.1 | 0.9 | 0.3 | 0.2 |
| Median absolute breakpoint size | 1.1 °C | 0.8°C | 1.1 °C | 0.8 °C | 1.25 | 1.15 |
| Mean absolute breakpoint size | 1.5 °C | 1.0 °C | 1.6 °C | 1.2 °C | 1.30 | 1.20 |
| Maximum absolute breakpoint size | 10.2 °C | 5.0 °C | 8.4 °C | 5.0 °C | 3.40 | 2.00 |

**Table 5: Trends of spatially averaged anomalies in the Altiplano (≥3500 m a.s.l.) in the period 1981 to 2010. Trends are shown for the annual means, for the 10th and 90th percentile of maximum temperature (TX) and minimum temperature (TN) (i.e. TX10p, TN10p, TX90p, TN90p), and for the number of frost days (FD). Bold numbers denote significance at the 5 % level.**

| | TX | | | | TN | | | |
|---|---|---|---|---|---|---|---|---|
| | DATA QC-S | DATA QC-S_H | DATA QC-E | DATA QC-E_H | DATA QC-S | DATA QC-S_H | DATA QC-E | DATA QC-E_H |
| Annual means (°C decade$^{-1}$) | **+0.41** | **+0.42** | **+0.44** | **+0.40** | -0.04 | +0.05 | -0.12 | **+0.22** |
| 10th percentile (days decade$^{-1}$) | **-13.2** | **-14.4** | **-12.0** | **-11.9** | +0.4 | - 1.0 | -0.9 | **-5.8** |
| 90th percentile (days decade$^{-1}$) | **+8.7** | **+11.0** | **+9.8** | **+9.3** | +0.2 | +3.7 | +2.0 | **+8.8** |
| FD (days decade$^{-1}$) | --- | --- | --- | --- | +2.9 | -1.3 | +1.4 | **-6.5** |

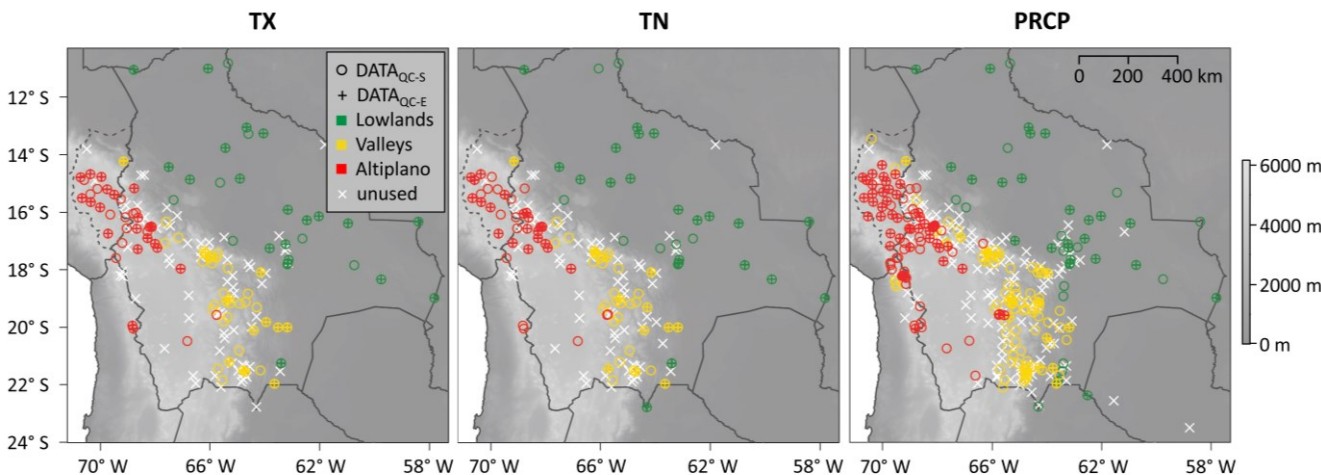

**Figure 1: Stations of the DECADE dataset with ≥20 years of valid observations for maximum temperature (TX), minimum temperature (TN), and precipitation (PRCP). Solid lines represent country boarders, and the dashed line the boarder of the Peruvian department of Puno. Circles and pluses indicate stations with ≥80 % of valid measurements 1981 to 2010 in the datasets quality**
10 **controlled with a standard method (DATA_QC-S) and with an enhanced approach (DATA_QC-E), respectively. White crosses mark stations with <80 % of valid observations 1981 to 2010 in both datasets. Colours classify stations regarding their elevation in Lowlands (≤600 m a.s.l.), Valleys (601 to 3499 m a.s.l.), and Altiplano (≥3500 m a.s.l.). The grey background shading indicates the elevation in m a.s.l.**

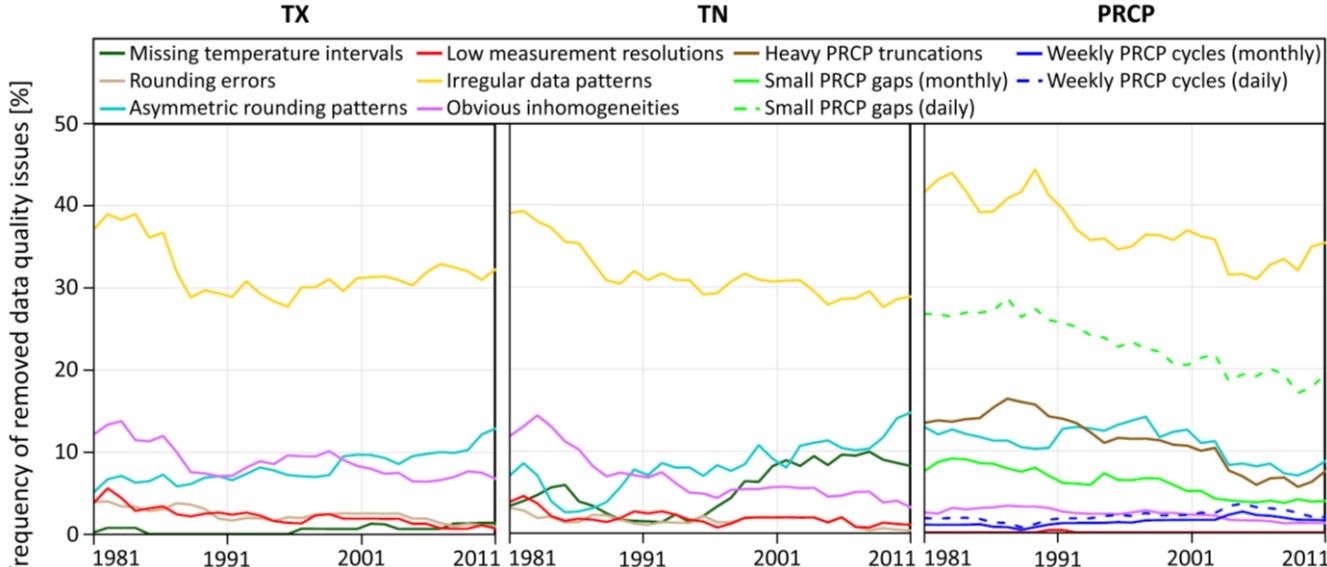

**Figure 2: Annual frequency of the data quality issues that cause the exclusion of the affected time series segments for maximum and minimum temperature (TX and TN, respectively), and precipitation (PRCP). If not specified, the frequencies apply to daily observations and monthly aggregations. Note that tests for systematic data quality issues were done in parallel, and time series segments may therefore be affected by several quality issues simultaneously.**

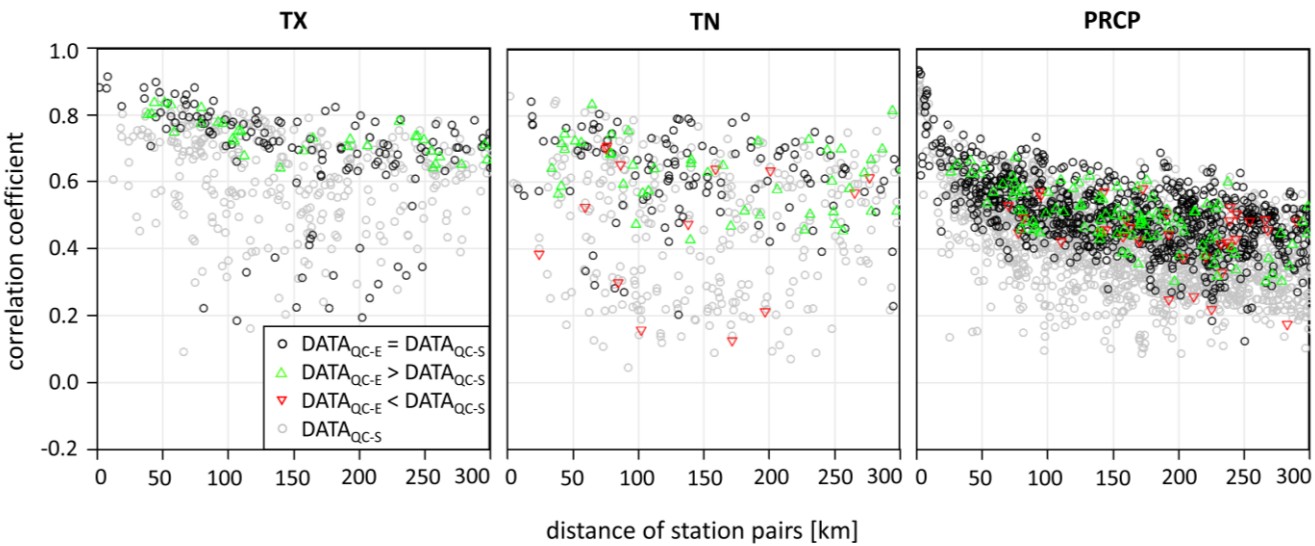

**Figure 3: Correlation coefficients of station pairs as function of station distance for maximum temperature (TX), minimum temperature (TN), and precipitation (PRCP). This figure shows the example of monthly correlations in the Altiplano (≥3500 m a.s.l.). Black circles indicate equal correlation coefficients in $DATA_{QC-S}$ and $DATA_{QC-E}$ (absolute difference ≤0.01), grey circles indicate correlation coefficients of station combinations of $DATA_{QC-S}$ that do not occur in $DATA_{QC-E}$ (or the absolute difference to the equivalent in $DATA_{QC-E}$ is >0.01), green triangles show correlation coefficients of $DATA_{QC-E}$ that are higher than in $DATA_{QC-S}$ (difference >+0.01), and red triangles show correlation coefficients of $DATA_{QC-E}$ that are lower than $DATA_{QC-S}$ (difference <-0.01).**

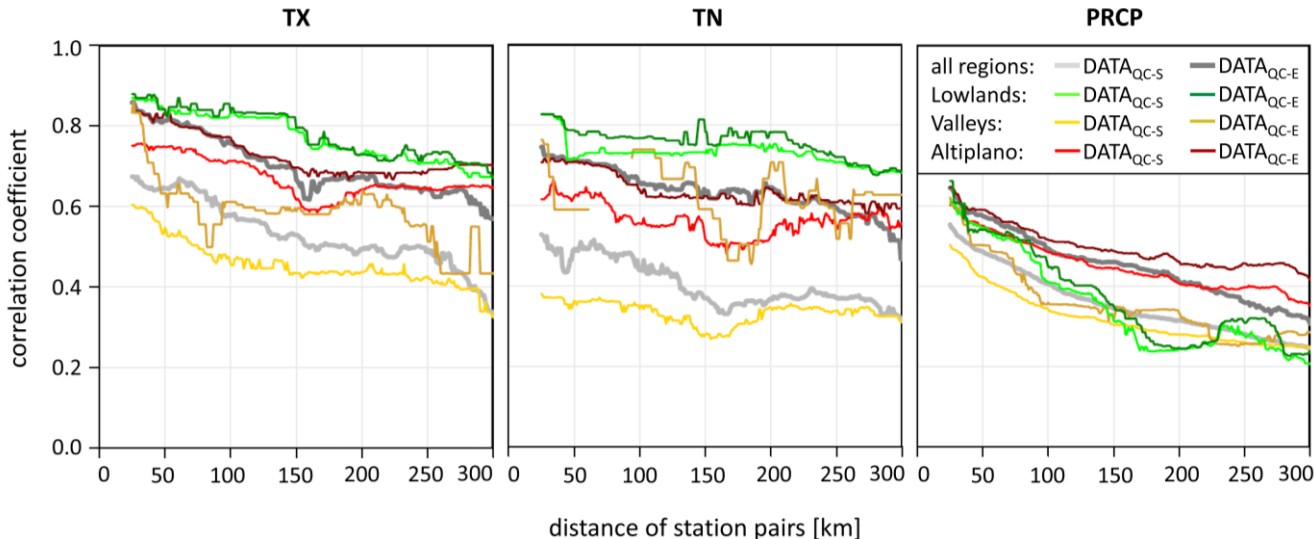

**Figure 4: Monthly median correlation coefficients within a 49-km running window for maximum temperature (TX), minimum temperature (TN), and precipitation (PRCP). The median correlation coefficient is not shown if there are less than three station pairs within the running window. Colours mark the medians for all regions combined, the Lowlands (≤600 m a.s.l.), the Valleys (601 to 3499 m a.s.l.), and the Altiplano (≥3500 m a.s.l.). Light and dark colours indicate correlation coefficients derived from DATA$_{QC-S}$ and DATA$_{QC-E}$, respectively.**

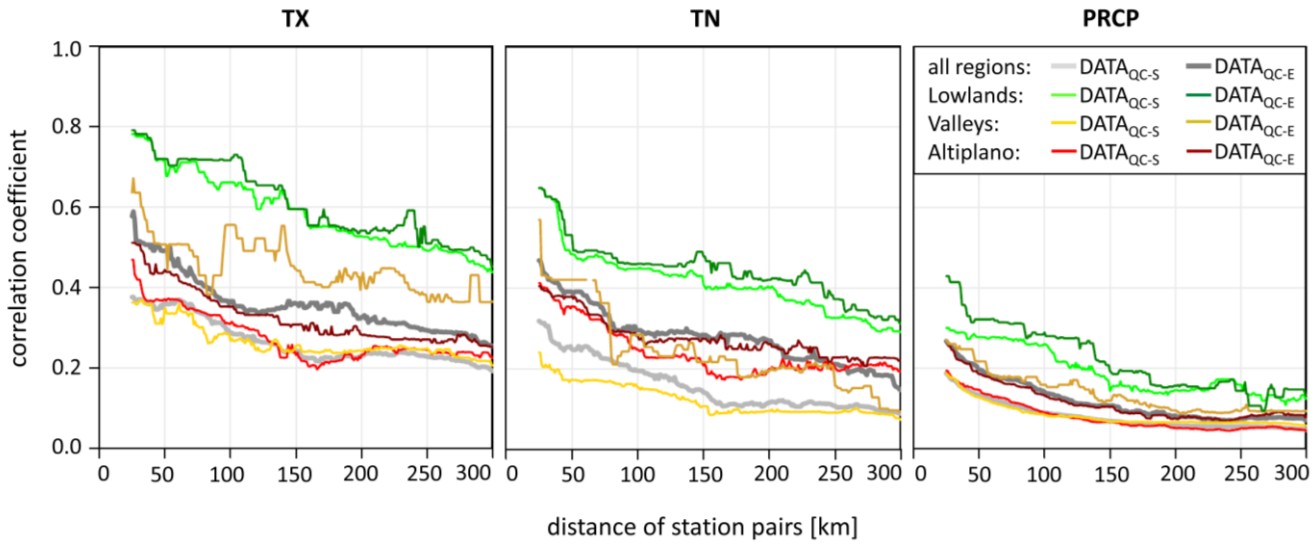

**Figure 5: Same as Fig. 4 but for daily data.**

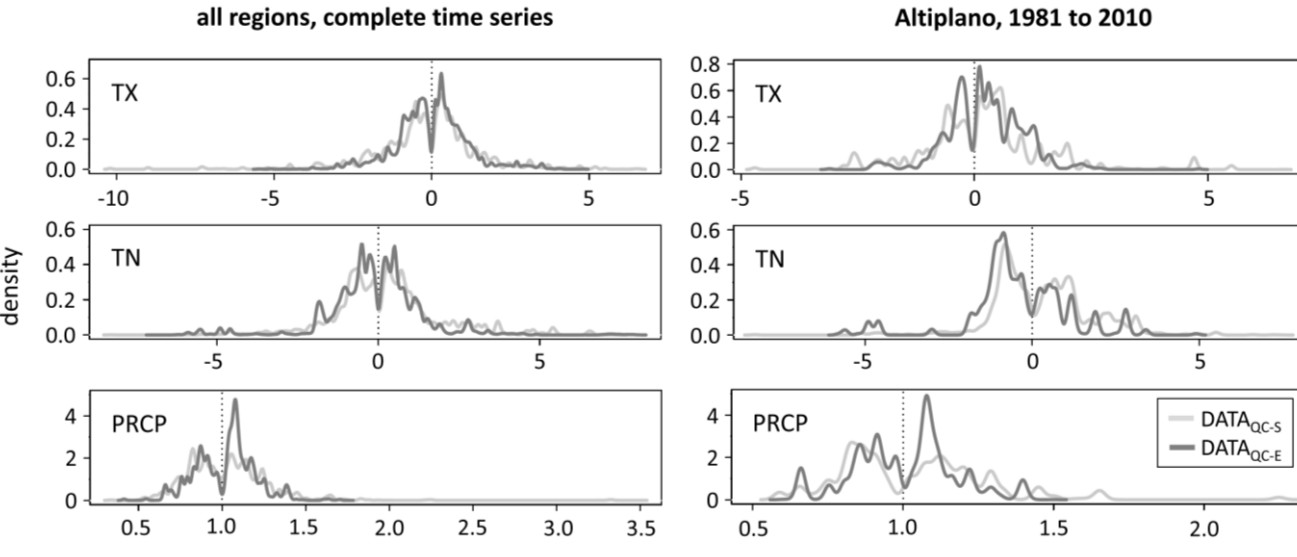

**Figure 6: Kernel density of the adjustments calculated with ACMANT3 for all regions and the complete time series (left plots), and for the Altiplano stations in 1981 to 2010 (right plots). For maximum and minimum temperature (TX and TN, respectively), inhomogeneous time series segments are corrected by adding the adjustment values, whereas for precipitation (PRCP), inhomogeneous segments are corrected by multiplication with the adjustment factors.**

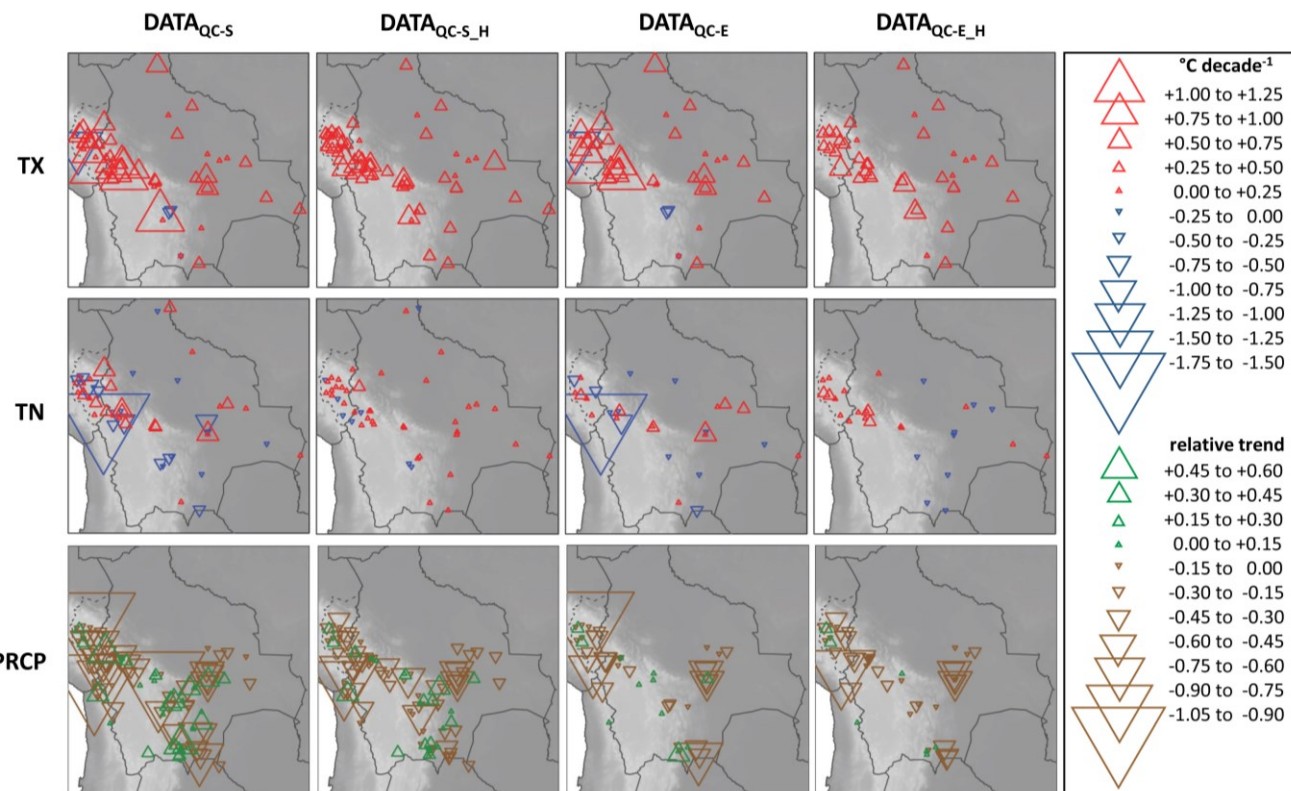

**Figure 7: Trends of individual station records for maximum temperature (TX) (first row), minimum temperature (TN) (second row), and precipitation (PRCP) (third row) 1981 to 2010. The first column shows the results for the unhomogenized dataset quality controlled with the standard approach (DATA$_{QC-S}$), the second column the homogenized dataset quality controlled with the standard approach (DATA$_{QC-S\_H}$), the third column the unhomogenized dataset quality controlled with the enhanced approach (DATA$_{QC-E}$), and the fourth column the homogenized dataset quality controlled with the enhanced approach (DATA$_{QC-E\_H}$). For temperature, trends are indicated in °C per decade. For PRCP, the relative magnitudes of the trend changes 1981 to 2010 are shown. It is calculated from the difference between the fitted value at the end and the beginning of the time series, which is divided by the mean of the fit. A relative trend increase by 1 is equal to an increase by 200.0 %, and a relative decrease by 1 is equal to a decrease by 66.7 %.**

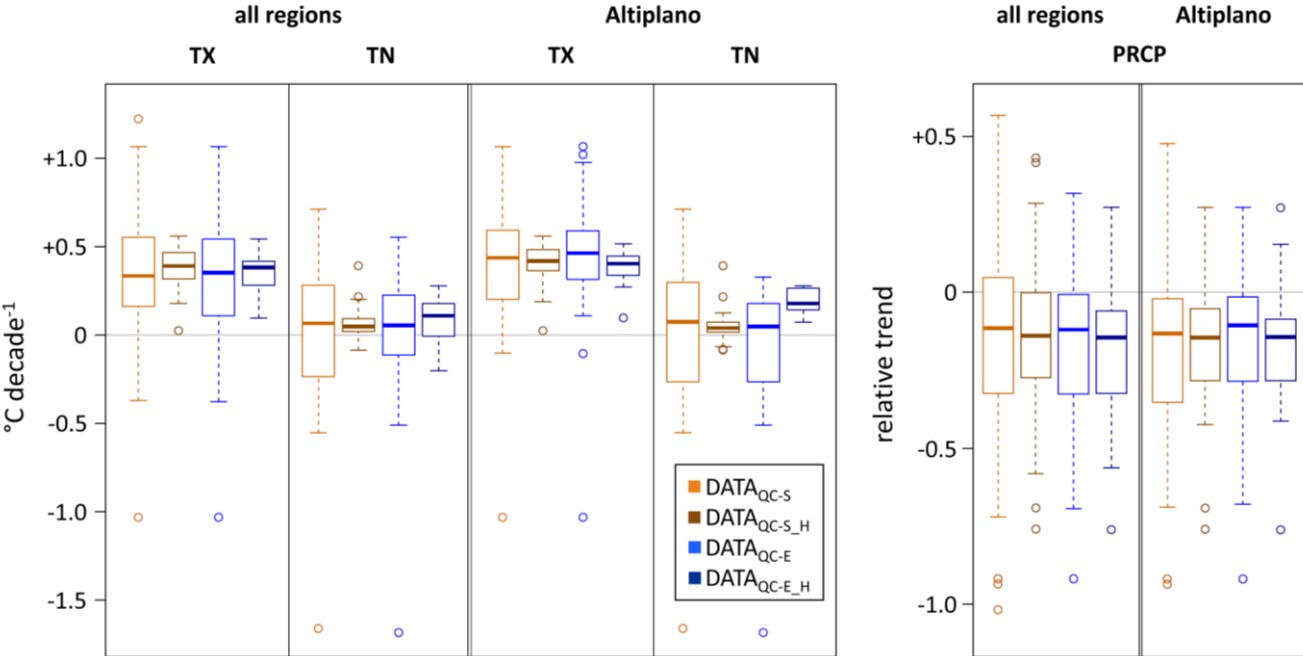

**Figure 8: Trends of individual station records for maximum temperature (TX), minimum temperature (TN), and precipitation (PRCP) in the period 1981 to 2010. Trend boxplots for the complete study area and for the Altiplano (≥3500 m a.s.l.) are shown. Colours indicated the different datasets that are unhomogenized and quality controlled with the standard approach (DATA_{QC-S}), homogenized and quality controlled with the standard approach (DATA_{QC-S_H}), unhomogenized and quality controlled with the enhanced approach (DATA_{QC-E}), and homogenized and quality controlled with the enhanced approach (DATA_{QC-E_H}). For temperature, trends are specified in °C per decade. For PRCP, relative trends in 1980 to 2010 are shown (see Caption of Fig. 7 for details). The boxplots show the median, the 25th and 75th percentile, and the 1.5×IQR (whiskers).**

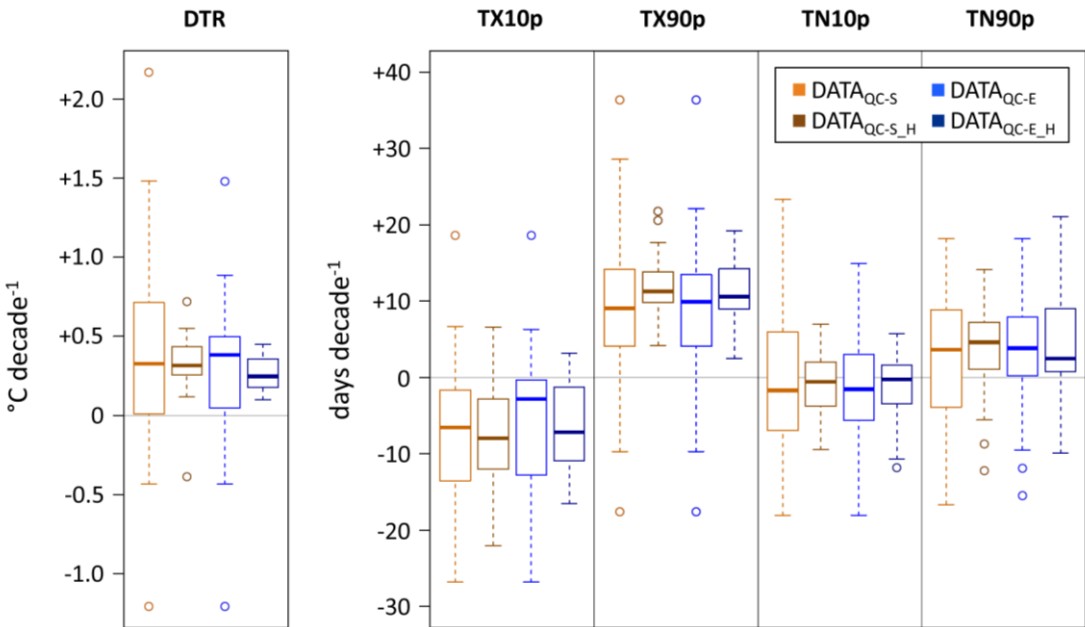

**Figure 9: Trends of individual station records of the complete study area for the climate change indices daily temperature range (DTR), number of cool days (TX10p), number of warm days (TX90p), number of cool nights (TN10p), and number of warm nights (TN90p) in the period 1981 to 2010. Colours indicated the different datasets that are unhomogenized and quality controlled with the standard approach (DATA$_{QC-S}$), homogenized and quality controlled with the standard approach (DATA$_{QC-S\_H}$), unhomogenized and quality controlled with the enhanced approach (DATA$_{QC-E}$), and homogenized and quality controlled with the enhanced approach (DATA$_{QC-E\_H}$). For the DTR, trends are specified in °C per decade, and for the other indices in days per decade. The boxplots show the median, the 25th and 75th percentile, and the 1.5×IQR (whiskers).**

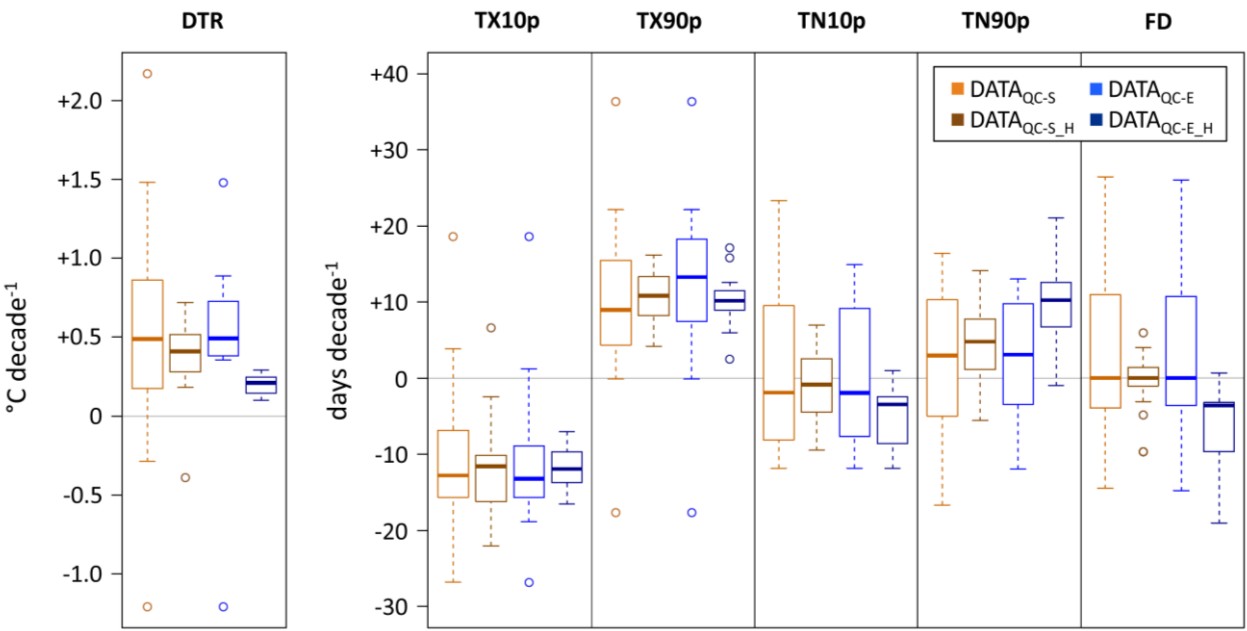

**Figure 10: Same as Fig. 9 but for the Altiplano stations (≥3500 m a.s.l.). Additionally, trends of frost days (FD) are shown.**