# Peer review of "Effects of undetected data quality issues on climatological analyses"

_Climate of the Past, 2017_

## Referee Comment (RC1) · Anonymous Referee #1 · 9 Jun 2017

Review of Hunziker et al., 'Effects of undetected data quality issues on climatological analyses' The issues raised in this paper are important – in particular, that data quality control methods which focus on daily data may not detect systematic issues that are important at annual or monthly timescales, and that those systematic issues may distort trends and/or homogenisation processes.

The key substantive result of this paper, in my view, was that for TN, the median adjustment in DATAQC-E was very different to that in DATAQC-S (with flow-on consequences for the trends). However, there is no indication in the paper as to why such a difference might have happened. One could reasonably form a null hypothesis that data quality issues might be expected to be randomly distributed in sign and in time; the results found suggest that there is a systematic departure from that null hypothesis, but without any

information as to what might be driving the difference, it is very hard to know how to interpret results (or whether they might be applicable to other networks). I think further analysis/discussion of the cause of this difference is important in a revised paper.

Other comments are as follows:

**Major comments**

P2 line 27 (and elsewhere) – the text at various points suggests that the observed increase in diurnal temperature range may be indicative of a problem. There are cases elsewhere in the world where increases in diurnal range have been associated with drying trends (e.g. Dittus et al 2014 (Aust. Met. Oceanogr. J. 64-261-71) found an association between recent decreases in rainfall and increased frost frequency in parts of southern Australia), and the diurnal temperature range increases found in this study seem to me to be broadly consistent with the observed decreases in precipitation. Potential rainfall-DTR relationships could also be brought into the discussion in sections 4.3.2 and 4.3.3, as well as being linked to the statement 'stronger increase in TX than TN in the Altiplano' at page 14 line 22.

P4 line 19-20 – does this mean that TX at these stations is not over a full 24 hours? If so, what is the time window that is used, and how confident are we that this practice has remained unchanged over the period of observations? Would TN also be over less than 24 hours? (if so, this might lead to low minimum temperatures which occur during afternoon storms – a common scenario in the tropics – being missed).

Section 3.1.2 – I found this section very difficult to understand as a standalone paper. While it is reasonable to refer people to the IJC paper for full details (I also note that it is an open-access paper), I think more explanation is needed in this section so that readers can have at least a basic understanding of what is happening without having to look up a different paper. A new table would be useful, I think, with a brief explanation of each test – at present, for example, it is not at all obvious to the reader what 'missing temperature interval' or 'PRCP gaps' mean. Some of the checks in this list also aren't
fully explained in the IJC paper (moderate and strong irregularities in the data pattern, frequent and large inhomogeneities, strong asymmetric rounding patterns).

Section 3.1.2 - I would also question how important the 'rounding errors' are. The example given in the IJC paper would affect temperatures by less than  $0.1^{\circ}C$  (which would not be worth worrying about), but there may be more significant examples.

Section 3.1.2 – flagging of 40% of measurements is a large number. It would be useful to have an indication of which individual tests most commonly led to flagging. With 40% of observations flagged, presumably a substantial number of stations were removed altogether – how many? (Somewhere – maybe section 3.4 – it would be useful to say how many of the original stations were still available for analysis for each of the two methods). It would also be useful to know whether UDQIs were concentrated in one particular era or spread fairly evenly through the time series.

Section 3.5 – it needs to be made clear that these trends are for the 1981-2010 period (as it is currently worded, there is the possibility of confusion between the timespan over which trends are calculated and the timespan used as a baseline for normal/index calculation).

P9 line 4-14 – the wording in this paragraph is not as clear as it might be. It is a perfectly reasonable decision to use a monthly method for comparative purposes, since what you are trying to do here is compare one QC method with another. (Also, as far as I know, there are no fully automated daily methods in existence).

Section 4.1 – it's probably also worth noting somewhere that TX correlations are much stronger than TN correlations in the valleys (presumably because TN is much more strongly influenced by local topography). Another question which may be worth considering is the extent to which there might be seasonal influences on correlations in a tropical climate – experience from other parts of the world suggest that TN correlation length scales in tropical climates are much shorter in the rainy season than in the dry season (perhaps because in tropical climates in the wet season, low temperatures
sometimes occur during rain events).

P13 line 19 (and onwards) – another possible effect of UDQIs is that they could also hide real inhomogeneities – either directly, or indirectly through adding noise to a time series (and thus reducing the signal-noise ratio).

Minor comments

- P2 line 14 should be 'lose significance'.
- P6 line 30 'Spearman' should be capitalised.

P7 line 9 – 'usually becomes slightly negative' – do you mean 'weaker'? I wouldn't expect a day shift to reduce a strong positive correlation to below 0.

P8 line 1 – should be 'performances'

P10 line  $30 - 10.2^{\circ}$ C seems very large – is this actually a data quality problem large enough to trigger homogeneity checks?

P11 line 23 – 'inflation of the trend spread' – this doesn't make sense as the trend spread is decreasing, not increasing. I guess you could use 'deflation', but I think just 'decrease' is fine.

P13 line 13 – should read 'Peruvian Andes and Switzerland'.

P15 line 5 – should read 'trends of a few climate change indices'

P15 line 7 - should read 'drawing of clear conclusions'

P15 line 16-18 – while missed observations of a few millimetres may have a negligible effect on monthly sums, they still affect indices which use number of rain days > 1 mm as a basis (e.g. SDII).

References - I note that the Gubler et al paper is now published.

---

## Referee Comment (RC2) · Anonymous Referee #2 · 14 Jun 2017

Review of *Effects of undetected data quality issues on climatological analyses* by Hunziker et al.

**General comments**

This paper uses data from the central Andes to illustrate the impact of systematic data issues on climate analyses. By applying standard and enhanced quality control procedures before data homogenisation, the study shows that systematic biases – which might not be detected using standard QC methods – can have a big impact on the analysis of temperature and rainfall trends.

This paper is a nice contribution to the field of climate data quality, and raises some important considerations that are often overlooked when large-scale climate trend analysis is conducted. The study is succinct and well written, and is a good complement to this group's previous paper (Hunziker et al. 2017).

Some suggestions for possible improvements to the paper are given below for the authors' consideration.

**Specific comments**

1. The main concern I have about the results from the study is how much data are removed in the enhanced QC procedure, and what impact that would have on the final trends. Around 40% is a significant amount of data, and you'd expect that removing that much information would have an effect on the homogenisation and analyses whether or not the data have issues.
   It'd be great if the authors could repeat their analysis using a synthetic dataset, or a dataset that does not have systematic data issues and see the impact of removing 40% of the data. That way they could ascribe some statistical significance to their findings. If this is not plausible, then at least a discussion on the role of missing data in the results, OR more detail on where and when the removed data are.
2. In section 3.1.2, I think a little more information is needed about the QC issues mentioned. I know it's in Hunziker et al. 2017, but the two papers are independent and should be understandable on their own.

**Technical corrections**

- Introduction line 2: replace "most" with "many national"
- Intro lines 20–30: This part confused me a little. Are you saying that trends actually do vary between neighbouring stations (due to climate factors), or that issues with data quality cause the differences?
- Page 3, line 1: as "the" enhanced approach
- Page 3, lines 5–9: The Central Andean region is also a good case study because of its complex terrain, as quality issues/homogenisation is notoriously difficult in such conditions.
- Page 3, lines 9–12: Change chapter to sections
- Page 3, line 20: add "network" after weather observation
- Page 4, line 5: data "were" quality-controlled
- Page 4, lines 10–15: I'd add that the Durre et al. tests include spatial inconsistency tests.

- Page 4, lines 27–29: Why doesn't the GHCN-Daily QC approach flag these clearly erroneous values?
- Page 4, line 30: add % after 0.35
- Page 5, line 6: on "an" annual time scale
- Page 5, line 26: add % after 0.26
- Page 5, lines 28 and 30 [and throughout the manuscript]: I think you need "a" before monthly and daily time scale.
- Page 6, line 5–6: I suggest you reword to: Note that Hunziker et al. (2017) further suggest *the inclusion of* additional information derived from 5 metadata into the QC process. This allows *the removal* station records that were generated under inappropriate conditions such as poor station siting or severe…
- Page 6, line 15: one day "a week"
- Page 6, line 30: I'd add reference to Peterson et al. 1998 when discussing using first-differences: Peterson, T. C., T. R. Karl, P. F. Jamason, R. Knight, and D. R. Easterling (1998), First difference method: Maximizing station density for the calculation of long-term global temperature change, J. Geophys. Res., 103(D20), 25967–25974, doi:10.1029/98JD01168.
- Page 6, line 31: Spearman (with a capital S)
- Page 7, line 1: add "the" before monthly time scale
- Section 3.4.1. Is there a way to graphically show the different clusters? Perhaps some additional panels in Figure 1 using polygons?
- Section 3.4.2. It sounds like you've used ACMANT3 because it is fully automated. Perhaps mention this earlier in the section to make that point stronger.
- Page 8, line 14: remove "of"
- Page 8, line 15: remove the hyphen between DECADE and dataset.
- Page 8, line 16: add "and" after requirements
- Page 8, lines 20–24: You need an extra line or two here to define/explain the Theil-Sen estimator, and how you have pre-whitened the data.
- Page 9, lines 5-15: I understand what you're trying to say here, but it's a bit confusing. Consider revising the text.
- Page 13, line 13: "and" Switzerland
- Page 13, line 14: Why are you hypothesising this? Why would the tropics have more UDQI? Do you also mean that there are more likely to be UDQI in developing countries?
- Page 15, line 5: add "a" before few climate change indices
- Table 1 caption: were, not where
- Table 2: can you add any information about the spread of the data in this table?
- Figure 2 caption: I'd add the word "show" after the words green triangles and red triangles.

---

## Editor Comment (EC1) · V. Rath (Editor) · 14 Jul 2017

Dear authors,

your manuscript has been received positively by both referees, though there are some critical comments. You should now reply in detail to these comments, and prepare a revised version of the manuscript accordingly.

Best regards,

Volker Rath

---

## Author Comment (AC1) · 9 Aug 2017

We want to thank the anonymous referee #1 for the important and helpful comments and suggestions.

Response to Anonymous Referee #1

Review of Hunziker et al., 'Effects of undetected data quality issues on climatological analyses' The issues raised in this paper are important – in particular, that data quality control methods which focus on daily data may not detect systematic issues that are important at annual or monthly timescales, and that those systematic issues may distort trends and/or homogenisation processes.

The key substantive result of this paper, in my view, was that for TN, the median adjustment in DATAQC-E was very different to that in DATAQC-S (with flow-on consequences for the trends). However, there is no indication in the paper as to why such a difference might have happened. One could reasonably form a null hypothesis that data quality issues might be expected to be randomly distributed in sign and in time; the results found suggest that there is a systematic departure from that null hypothesis, but without any information as to what might be driving the difference, it is very hard to know how to interpret results (or whether they might be applicable to other networks). I think further analysis/discussion of the cause of this difference is important in a revised paper.

Response 1: We agree with the referee that investigating the sources of the differences in the correction factor for TN for DATAQC-S and DATAQC-E would be an important further step. However, finding such explanations would require much further analyses (mostly in-situ), particularly in a sparse network with very limited metadata availability. Reconstructing the station history is nearly impossible in many cases. Therefore, potential sources causing a warm bias in former TN observations cannot be investigated in detail in this paper. Nevertheless, we will include a discussion of this important issue in the article. Furthermore, we will analyze if certain data quality issues occur more/less frequently in recent years and decades and if/how this may impact the adjustments in data homogenization.

Other comments are as follows:

Major comments

P2 line 27 (and elsewhere) – the text at various points suggests that the observed increase in diurnal temperature range may be indicative of a problem. There are cases elsewhere in the world where increases in diurnal range have been associated with drying trends (e.g. Dittus et al 2014 (Aust. Met. Oceanogr. J. 64-261-71) found an association between recent decreases in rainfall and increased frost frequency in parts of southern Australia), and the diurnal temperature range increases found in this study seem to me to be broadly consistent with the observed decreases in precipitation.

Potential rainfall-DTR relationships could also be brought into the discussion in sections 4.3.2 and 4.3.3, as well as being linked to the statement 'stronger increase in TX than TN in the Altiplano' at page 14 line 22.

Response 2: We will discuss this question in more detail in the paper. We will reference the publication of Dittus et al. (2014) and other relevant works. Such rainfall-DTR relationships may potentially be an explanation for the increasing DTR trend signal that still remains after an enhanced QC. The statement in the introduction does not imply that increasing DTR is necessarily the consequence of data quality problems. But the findings in earlier studies draw the attention to regions such as the Central Andean area and motivate to further investigate the causes of the detected trends (climatological or non-climatological). In-depth investigations of potential rainfall-DTR relationships is out of scale for the present publication but would definitely be an interesting research question for a follow-up publication.

P4 line 19-20 – does this mean that TX at these stations is not over a full 24 hours? If so, what is the time window that is used, and how confident are we that this practice has remained unchanged over the period of observations? Would TN also be over less than 24 hours? (if so, this might lead to low minimum temperatures which occur during afternoon storms – a common scenario in the tropics – being missed).

Response 3: We will add some information to these lines to make the issue more clear. At most stations, TX measurements do not cover 24 hours indeed. TX may only cover the afternoon hours (ca. 1200 to 1800 LST), when daily maximum temperatures normally occur. In certain cases however, this may lead to too low daily TX measurements. For TN, however, we are not aware of such practices, and we expect TN to be representative for 24 hours (measurements are normally taken at 0800 LST). To our knowledge, there were no systematic network wide changes in these observation practices. Because of the very limited information about the station histories, different and changing observation practices are possible at individual stations however.

Section 3.1.2 – I found this section very difficult to understand as a standalone paper. While it is reasonable to refer people to the IJC paper for full details (I also note that it is an open-access paper), I think more explanation is needed in this section so that readers can have at least a basic understanding of what is happening without having to look up a different paper. A new table would be useful, I think, with a brief explanation of each test – at present, for example, it is not at all obvious to the reader what 'missing temperature interval' or 'PRCP gaps' mean. Some of the checks in this list also aren't fully explained in the IJC paper (moderate and strong irregularities in the data pattern, frequent and large inhomogeneities, strong asymmetric rounding patterns).

Response 4: The referee is right that the tests of this work cannot be found completely in the previous IJC publication of Hunziker et al. (2017). We also agree that the present paper should be understandable without reading another publication. Therefore, we will include a table that summarizes all the tests mentioned in this paper. The explanations will be concise but they will make clear what kind of errors were detected in the dataset. The IJC publication of Hunziker et al. (2017) should just serve to find more details on the data quality issues.

Section 3.1.2 – I would also question how important the 'rounding errors' are. The example given in the IJC paper would affect temperatures by less than 0.1_C (which would not be worth worrying about), but there may be more significant examples.

Response 5: The effects of the rounding errors from degrees Fahrenheit to degrees Celsius are indeed not highly important. However, in these cases there is an accumulation of uncertainties: 1) the observer was not aware from which scale to read the temperature (in most cases, the instruments triggering the error have scales of the two units), and 2), we do not know why and how exactly the rounding error from degrees Fahrenheit to degrees Celsius occurred. Therefore, there are potentially other errors in those measurements, and it seems reasonable to not use such data for climatological analysis.

Section 3.1.2 – flagging of 40% of measurements is a large number. It would be useful to have an indication of which individual tests most commonly led to flagging. With 40% of observations flagged, presumably a substantial number of stations were removed altogether – how many? (Somewhere – maybe section 3.4 – it would be useful to say how many of the original stations were still available for analysis for each of the two methods). It would also be useful to know whether UDQIs were concentrated in one particular era or spread fairly evenly through the time series.

Response 6: We will add additional information on the frequency of the flagging by the tests applied, potentially included in the table described in Response 4. We will also include information on the temporal occurrence of data quality issues. The number of available stations varies between the different analyses. We will provide the most important numbers of available time series for the analyses and the differences resulting from the two data quality control approaches.

Section 3.5 – it needs to be made clear that these trends are for the 1981-2010 period (as it is currently worded, there is the possibility of confusion between the timespan over which trends are calculated and the timespan used as a baseline for normal/index calculation).

Response 7: The wording will be adapted to avoid any confusion between the time period over which trends are calculated and the baseline period for the index calculation. Since we are calculating trends for the 30-year period 1981-2010, the required 30-year baseline period for the index calculation is the same.

P9 line 4-14 – the wording in this paragraph is not as clear as it might be. It is a perfectly reasonable decision to use a monthly method for comparative purposes, since what you are trying to do here is compare one QC method with another. (Also, as far as I know, there are no fully automated daily methods in existence).

Response 8: We will revise the wording of this paragraph.

Section 4.1 – it's probably also worth noting somewhere that TX correlations are much stronger than TN correlations in the valleys (presumably because TN is much more strongly influenced by local topography). Another question which may be worth considering is the extent to which there might be seasonal influences on correlations in a tropical climate – experience from other parts of the world suggest that TN correlation length scales in tropical climates are much shorter in the rainy season than in the dry season (perhaps because in tropical climates in the wet season, low temperatures sometimes occur during rain events).

Response 9: We will include some sentences on the differences between the correlation of TX and TN within the different regions. We agree with the referee that there are likely differences between the correlations detected in the different seasons. However, since the present paper investigates the differences resulting from different QC approaches and not mainly the climatology in the Central Andean area, we consider the seasonal scale of correlations (as well as of trends) an aspect of minor importance. Regarding the length and the large amount of information of the present paper, we prefer to not include analyses on seasonal time scale.

P13 line 19 (and onwards) – another possible effect of UDQIs is that they could also hide real inhomogeneities – either directly, or indirectly through adding noise to a time series (and thus reducing the signal-noise ratio).

Response 10: We will include this important aspect in the discussion.

Minor comments

P2 line 14 – should be 'lose significance'.

Response 11: Will be corrected.

P6 line 30 – 'Spearman' should be capitalised.

Response 12: Will be corrected.

P7 line 9 – 'usually becomes slightly negative' – do you mean 'weaker'? I wouldn't expect a day shift to reduce a strong positive correlation to below 0.

Response 13: Shifting two highly correlated time series by one day indeed often results in a slightly negative correlation of the first differences in this tropical area.

P8 line 1 – should be 'performances'

Response 14: Will be corrected.

P10 line 30 – 10.2_C seems very large – is this actually a data quality problem large enough to trigger homogeneity checks?

Response 15: Data quality issues often trigger the statistical detection of inhomogeneities. This very large adjustment did occur in the time series shown below (Fig. 1) between the segments 1990-1994 and 1997-2007. This time series is (at least temporarily) affected by several systematic data quality issues: strong asymmetric rounding patterns (e.g. around 2000), moderate and strong irregularities in the data patterns (not clearly classifiable but highly suspicious patterns, such as strong changes in the variance, truncations, or missing temperature ranges), and frequent and large inhomogeneities. The low quality of the data suggests that the inhomogeneity rather originates from an observer error (e.g. erroneous reading of the instrument) than from other impacts (e.g. station relocation).

P11 line 23 – 'inflation of the trend spread' – this doesn't make sense as the trend spread is decreasing, not increasing. I guess you could use 'deflation', but I think just 'decrease' is fine.

Response 16: Will be corrected.

P13 line 13 – should read 'Peruvian Andes and Switzerland'.

Response 17: Will be corrected.

P15 line 5 – should read 'trends of a few climate change indices'

Response 18: Will be corrected.

P15 line 7 – should read 'drawing of clear conclusions'

Response 19: Will be corrected.

P15 line 16-18 – while missed observations of a few millimetres may have a negligible effect on monthly sums, they still affect indices which use number of rain days > 1 mm as a basis (e.g. SDII).

Response 20: We will reformulate the sentence. Here, we want to stress the fact that UDQI may not have the same effect in different regions. Missing observations of small PRCP events (e.g. up to 2 mm) may not affect the monthly precipitation sums in a wet region, but in a dry area these missed observations may bias monthly PRCP sums.

References – I note that the Gubler et al paper is now published.

[Figure]

**Fig. 1.** Time series for which the largest temperature adjustment values were calculated in the data homogenization process.

---

## Author Comment (AC2) · 9 Aug 2017

We want to thank the anonymous referee #2 for the important and helpful comments and suggestions.

Anonymous Referee #2

General comments

This paper uses data from the central Andes to illustrate the impact of systematic data issues on climate analyses. By applying standard and enhanced quality control procedures before data homogenisation, the study shows that systematic biases – which might not be detected using standard QC methods – can have a big impact on the analysis of temperature and rainfall trends.

[Figure]

This paper is a nice contribution to the field of climate data quality, and raises some important considerations that are often overlooked when large-scale climate trend analysis is conducted. The study is succinct and well written, and is a good complement to this group's previous paper (Hunziker et al. 2017).

Some suggestions for possible improvements to the paper are given below for the authors' consideration.

Specific comments

1. The main concern I have about the results from the study is how much data are removed in the enhanced QC procedure, and what impact that would have on the final trends. Around 40% is a significant amount of data, and you'd expect that removing that much information would have an effect on the homogenisation and analyses whether or not the data have issues. It'd be great if the authors could repeat their analysis using a synthetic dataset, or a dataset that does not have systematic data issues and see the impact of removing 40% of the data. That way they could ascribe some statistical significance to their findings. If this is not plausible, then at least a discussion on the role of missing data in the results, OR more detail on where and when the removed data are.

Response 1: We agree with the referee that 40% of the data is a relatively significant amount of information. The effect of removing 40% of the data, however, will strongly differ between different datasets. Furthermore, the results may differ significantly depending on which time series segments are (randomly) excluded. In the authors' opinion, the best approach would be to resample all the available station data quality controlled with the standard QC approach by removing 40% of the data randomly. Repeating this process many times, the resulting climatological analyses from these random groups may provide information about the statistical significance of the results of the present paper. However, the process from the dataset to the climatological analyses is time consuming (clustering, data homogenization, data analyses).

Therefore, such an approach is hardly feasible in frame of the present study. Two main evidences make it very unlikely that the differences found in the present study are just an effect of removing 40% of the data: 1) the enhanced QC removes obvious observation errors that cannot be corrected by data homogenization and consequently affect climatological analyses, and 2) the results of the dataset quality controlled with the enhanced approach are highly consistent (e.g., apparently better performance of data homogenization approach, more consistent station trends, low trend spread of the diurnal temperature range which indicates that the individual data homogenization of TX and TN are in accordance). We will add more information on the availability of time series for the different analyses (for the standard and the enhanced QC approach), and the frequency and temporality of the occurrence of the different data quality issues.

2. In section 3.1.2, I think a little more information is needed about the QC issues mentioned. I know it's in Hunziker et al. 2017, but the two papers are independent and should be understandable on their own.

Response 2: We agree with the referee. We will add information on the data quality issues in form of a table to make the paper fully understandable without reading the IJC publication of Hunziker et al. (2017).

Technical corrections

Introduction line 2: replace "most" with "many national"

Response 3: Will be adapted.

Intro lines 20–30: This part confused me a little. Are you saying that trends actually do vary between neighbouring stations (due to climate factors), or that issues with data quality cause the differences?

Response 4: We will reformulate this paragraph to make it clearer. The observation of varying trends between neighboring stations was a motivation to investigate if UDQI can cause or increase such inconsistencies. Later in our paper, we show that UDQI

indeed may increase spatial trend inconsistencies (Figure 6 in the paper).

Page 3, line 1: as "the" enhanced approach

Response 5: Will be corrected.

Page 3, lines 5–9: The Central Andean region is also a good case study because of its complex terrain, as quality issues/homogenisation is notoriously difficult in such conditions.

Response 6: We will include this aspect.

Page 3, lines 9–12: Change chapter to sections

Response 7: Will be adapted.

Page 3, line 20: add "network" after weather observation

Response 8: Will be corrected.

Page 4, line 5: data "were" quality-controlled

Response 9: Will be corrected.

Page 4, lines 10–15: I'd add that the Durre et al. tests include spatial inconsistency tests.

Response 10: Will be included.

Page 4, lines 27–29: Why doesn't the GHCN-Daily QC approach flag these clearly erroneous values?

Response 11: The reason for the problem is that some extremely high and low numbers did not fit in the maximum allowable number of digits in the GHCN-Daily data format. We will better explain the problem in the paragraph.

Page 4, line 30: add % after 0.35

Response 12: Will be corrected.

Page 5, line 6: on "an" annual time scale

Response 13: Will be corrected.

Page 5, line 26: add % after 0.26

Response 14: Will be corrected.

Page 5, lines 28 and 30 [and throughout the manuscript]: I think you need "a" before monthly and daily time scale.

Response 15: Will be corrected.

Page 6, line 5–6: I suggest you reword to: Note that Hunziker et al. (2017) further suggest the inclusion of additional information derived from 5 metadata into the QC process. This allows the removal station records that were generated under inappropriate conditions such as poor station siting or severe. . .

Response 16: Will be adapted.

Page 6, line 15: one day "a week"

Response 17: Will be corrected.

Page 6, line 30: I'd add reference to Peterson et al. 1998 when discussing using firstdifferences: Peterson, T. C., T. R. Karl, P. F. Jamason, R. Knight, and D. R. Easterling (1998), First difference method: Maximizing station density for the calculation of long-term global temperature change, J. Geophys. Res., 103(D20), 25967–25974, doi:10.1029/98JD01168.

Response 18: We will include the suggested reference.

Page 6, line 31: Spearman (with a capital S)

Response 19: Will be corrected.

Page 7, line 1: add "the" before monthly time scale

Response 20: Will be corrected.

Section 3.4.1. Is there a way to graphically show the different clusters? Perhaps some additional panels in Figure 1 using polygons?

Response 21: The clusters differ for each parameter and for the different QC approaches. Therefore, they cannot be marked in Figure 1. We will include a summarizing figure showing the different clusters in the supplementary material.

Section 3.4.2. It sounds like you've used ACMANT3 because it is fully automated. Perhaps mention this earlier in the section to make that point stronger.

Response 22: We may adapt the paragraph to make the point stronger as suggested.

Page 8, line 14: remove "of"

Response 23: Will be corrected.

Page 8, line 15: remove the hyphen between DECADE and dataset.

Response 24: Will be corrected.

Page 8, line 16: add "and" after requirements

Response 25: Will be corrected.

Page 8, lines 20–24: You need an extra line or two here to define/explain the Theil-Sen estimator, and how you have pre-whitened the data.

Response 26: We will include the required information.

Page 9, lines 5-15: I understand what you're trying to say here, but it's a bit confusing. Consider revising the text.

Response 27: We will revise the paragraph.

Page 13, line 13: "and" Switzerland

Response 28: Will be corrected.

Page 13, line 14: Why are you hypothesising this? Why would the tropics have more UDQI? Do you also mean that there are more likely to be UDQI in developing countries?

Response 29: Weather observation in less developed countries is at a different stage than weather observation in industrialized countries. Therefore, it is likely that more data quality issues occur in data from less developed countries that are particularly located in the tropics. We will revise these sentences.

Page 15, line 5: add "a" before few climate change indices

Response 30: Will be corrected.

Table 1 caption: were, not where

Response 31: Will be corrected.

Table 2: can you add any information about the spread of the data in this table?

Response 32: We may include the standard deviation to each median.

Figure 2 caption: I'd add the word "show" after the words green triangles and red triangles.

Response 33: Will be corrected.

---

## Author Response (AR1)

We want to thank the anonymous referee #1 for the important and helpful comments and suggestions.

**Response to Anonymous Referee #1**

5 *Review of Hunziker et al., 'Effects of undetected data quality issues on climatological analyses' The issues raised in this paper are important – in particular, that data quality control methods which focus on daily data may not detect systematic issues that are important at annual or monthly timescales, and that those systematic issues may distort trends and/or homogenisation processes.*

*The key substantive result of this paper, in my view, was that for TN, the median adjustment in DATAQC-E was very*
10 *different to that in DATAQC-S (with flow-on consequences for the trends). However, there is no indication in the paper as to why such a difference might have happened. One could reasonably form a null hypothesis that data quality issues might be expected to be randomly distributed in sign and in time; the results found suggest that there is a systematic departure from that null hypothesis, but without any information as to what might be driving the difference, it is very hard to know how to interpret results (or whether they might be applicable to other networks). I think further*
15 *analysis/discussion of the cause of this difference is important in a revised paper.*

Response 1: We did not get clear indication that the difference of the median adjustments in data homogenization process was introduced by UDQI directly. We provide more information about that by the new Table 1 and Figure 2. Hence, we expect that this difference occurs due to the noise introduced by UDQI impeding the detection of inhomogeneities of other sources (e.g. systematic changes in station siting or instruments). However, investigating such sources of inhomogeneities would
20 require much further analyses (mostly in-situ), particularly in a sparse network with very limited metadata availability. Reconstructing the station history is nearly impossible in many cases in the Central Andean area. Therefore, potential sources causing a warm bias in former TN observations in the Altiplano cannot be investigated in detail in this paper.

In order to make this issue clearer, we adapted the relevant paragraph in the Discussion as follows:

"The effect of UDQI also manifests in the adjustment values (temperature) and factors (PRCP) resulting from the data
25 homogenization process. UDQI cause a reduction in the frequency of small adjustments and an increase of large adjustments. They also may induce an adjustment bias. For instance, the median adjustment value for TN station records in the Altiplano is -0.5 °C. If the same dataset contains UDQI, the resulting median adjustment is +0.2 °C. This difference of the adjustment could be caused in two ways. First, UDQI may introduce a systematic bias (a cold bias in earlier observations in this case). This would require the occurrence of certain types of UDQI in many station records of a dataset which would cause a systematic
30 bias and which would strongly change their frequency in time. For the Central Andean area, however, there is no clear indication that UDQI meet these requirements in the period 1981 to 2010. Second, UDQI may not introduce a bias by themselves, but they impede the detection of an existing bias (warm bias in earlier observation in case of TN in the Altiplano) by introducing artificial noise. Such a warm bias could have been introduced, for example, by location changes of weather stations to systematically different sites (e.g. further away from buildings). The second possibility seems to be the more likely
35 cause of the observed adjustment differences of TN records in the Altiplano. Hence, UDQI may impede the adjustments of systematic biases introduced by inhomogeneities. In summary, UDQI may substantially decrease the performance of statistical data homogenization methods."

*Other comments are as follows:*

*Major comments*

*P2 line 27 (and elsewhere) – the text at various points suggests that the observed increase in diurnal temperature range may be indicative of a problem. There are cases elsewhere in the world where increases in diurnal range have*
5 *been associated with drying trends (e.g. Dittus et al 2014 (Aust. Met. Oceanogr. J. 64-261-71) found an association between recent decreases in rainfall and increased frost frequency in parts of southern Australia), and the diurnal temperature range increases found in this study seem to me to be broadly consistent with the observed decreases in precipitation. Potential rainfall-DTR relationships could also be brought into the discussion in sections 4.3.2 and 4.3.3, as well as being linked to the statement 'stronger increase in TX than TN in the Altiplano' at page 14 line 22.*

10 Response 2: We thank the reviewer for this interesting input. Also for the dataset free of UDQI, a positive DTR remains. We therefore added the following sentences to the relevant paragraph in the Discussion:

"An indication for a climatological explanation of the positive DTR trends are the simultaneously observed negative PRCP trends. Several authors have described a negative correlation between DTR and PRCP trends (Dittus et al., 2014; Jaswal et al., 2016; Zhou et al., 2009). Hence, the observations in the Altiplano would be in accordance with these findings."

15 *P4 line 19-20 – does this mean that TX at these stations is not over a full 24 hours? If so, what is the time window that is used, and how confident are we that this practice has remained unchanged over the period of observations? Would TN also be over less than 24 hours? (if so, this might lead to low minimum temperatures which occur during afternoon storms – a common scenario in the tropics – being missed).*

Response 3: At many stations in the Central Andean area, TX measurements do not cover 24 hours indeed. We adapted the
20 relevant paragraph as follows:

"However, the detections (i.e. the flags for failing certain tests) of the GHCN-Daily QC had to be slightly adapted in order to be more appropriate for weather observations in the Central Andean region. One of the internal consistency tests detects cases where TX is lower than TN of the previous day. This test should guarantee the physical consistency of TX and TN measurements that are representative for a 24-h period. However, in various Bolivian stations (particularly stations at airports),
25 TX is representative for the afternoon hours only (observations start at noon and end in the evening). This measurement practice should avoid problems in attributing the observation to a specific calendar day. Usually, daily temperature maxima occur in the afternoon indeed. Nevertheless, during certain weather events (particularly the frequent cold surges in the Lowlands (e.g. Espinoza et al., 2013; Garreaud, 2001; Vera and Vigliarolo, 2000)), the temperature in the afternoon does not exceed the TN value measured in the morning. As a result, a high number of observations in the Lowlands was flagged. To the authors'
30 knowledge, this measurement practice has been applied to TX but not to TN observations, and no large scale changes of this practice in the Central Andean area are known. Therefore, this practice (even though not ideal) does not introduce any error or bias as long as it remains unchanged. As a consequence, internal consistency flags set because of this particular QC test were regarded as invalid."

*Section 3.1.2 – I found this section very difficult to understand as a standalone paper. While it is reasonable to refer*
35 *people to the IJC paper for full details (I also note that it is an open-access paper), I think more explanation is needed in this section so that readers can have at least a basic understanding of what is happening without having to look up a different paper. A new table would be useful, I think, with a brief explanation of each test – at present, for example,*

*it is not at all obvious to the reader what 'missing temperature interval' or 'PRCP gaps' mean. Some of the checks in this list also aren't fully explained in the IJC paper (moderate and strong irregularities in the data pattern, frequent and large inhomogeneities, strong asymmetric rounding patterns).*

Response 4: The referee is right that the tests of this work cannot be found completely in the previous IJC publication of Hunziker et al. (2017). We also agree that the present paper should be understandable without reading another publication. As suggested by the reviewer, we included a table summarizing the data quality issues (Table 1) to the revise version.

*Section 3.1.2 – I would also question how important the 'rounding errors' are. The example given in the IJC paper would affect temperatures by less than 0.1_C (which would not be worth worrying about), but there may be more significant examples.*

Response 5: The effects of the rounding errors from degrees Fahrenheit to degrees Celsius are indeed not highly important. However, in these cases there is an accumulation of uncertainties: 1) the observer was not aware from which scale to read the temperature (in most cases, the instruments triggering the error have scales of the two units), and 2), we do not know why and how exactly the rounding error from degrees Fahrenheit to degrees Celsius occurred. Therefore, there are potentially other errors in those measurements, and it seems reasonable to not use such data for climatological analysis. In the new Table 1, we consider the reviewer's comment in the description of the data quality issue:

"Rounding errors in the conversion from degrees Fahrenheit to degrees Celsius (may also indicate further errors in the data)"

*Section 3.1.2 – flagging of 40% of measurements is a large number. It would be useful to have an indication of which individual tests most commonly led to flagging. With 40% of observations flagged, presumably a substantial number of stations were removed altogether – how many? (Somewhere – maybe section 3.4 – it would be useful to say how many of the original stations were still available for analysis for each of the two methods). It would also be useful to know whether UDQIs were concentrated in one particular era or spread fairly evenly through the time series.*

Response 6: Information on the spatial and temporal occurrence of the data quality issues was included in the revised version in the new Table 1 and Figure 2.

We also included the number of remaining station records for the most important climatological analyses, such as for correlation:

"Removing the flagged observations and time series without sufficient data resulted in 98 (TX), 99 (TN), and 218 (PRCP) valid monthly station records for DATAQC-S, and in 56 (TX), 54 (TN), and 105 (PRCP) valid monthly time series for DATAQC-E."

*Section 3.5 – it needs to be made clear that these trends are for the 1981-2010 period (as it is currently worded, there is the possibility of confusion between the timespan over which trends are calculated and the timespan used as a baseline for normal/index calculation).*

Response 7: We adapted the wording of the paragraph, most importantly:

"In order to investigate the effect of undetected data quality issues on extremes, we computed the frequently used climate change indices defined by the CCl/CLIVAR/JCOMM Expert Team on Climate Change Detection and Indices (ETCCDI) (http://etccdi.pacificclimate.org/list_27_indices.shtml) for 1981 to 2010.

…

5    For indices based on percentiles, the baseline period was calculated from the 30-year period 1981 to 2010."

*P9 line 4-14 – the wording in this paragraph is not as clear as it might be. It is a perfectly reasonable decision to use a monthly method for comparative purposes, since what you are trying to do here is compare one QC method with another. (Also, as far as I know, there are no fully automated daily methods in existence).*

Response 8: We adapted the paragraph as follows:

10    "The ETCCDI climate change indices describe moderate to very moderate extreme events that occur usually many times per year. Therefore, they are particularly suitable for the application on short time series. For the index calculation of the homogenized datasets, daily measurements were corrected by adding monthly adjustment values (temperature) and by multiplying with monthly adjustment factors (PRCP) that were computed with ACMANT3. Applying monthly corrections on a time series does not guarantee homogeneity on a daily time scale (Brönnimann, 2015; Costa and Soares, 2009; Trewin,
15    2013). However, since the present study aims to compare the effects of different QC methods, potential deficits in adjusting daily observations with monthly factors do not bias the results. Considering the large and frequent inhomogeneities detected in the Central Andean time series (Sect. 4.3), the homogeneity of the ETCCDI climate change indices will most likely be increased strongly by correcting the daily time series with the monthly adjustment values."

*Section 4.1 – it's probably also worth noting somewhere that TX correlations are much stronger than TN correlations*
20    *in the valleys (presumably because TN is much more strongly influenced by local topography). Another question which may be worth considering is the extent to which there might be seasonal influences on correlations in a tropical climate – experience from other parts of the world suggest that TN correlation length scales in tropical climates are much shorter in the rainy season than in the dry season (perhaps because in tropical climates in the wet season, low temperatures sometimes occur during rain events).*

25    Response 9: We agree with the referee that there are likely differences between the correlations detected in the different seasons. However, since the present paper investigates the differences resulting from different QC approaches and not mainly the climatology in the Central Andean area, we consider the seasonal scale of correlations (as well as of trends) an aspect of minor importance. Regarding the growing length and the large amount of information of the present paper, we prefer to not include analyses on seasonal time scale nor further findings on differences between TX and TN correlation.

30    *P13 line 19 (and onwards) – another possible effect of UDQIs is that they could also hide real inhomogeneities – either directly, or indirectly through adding noise to a time series (and thus reducing the signal-noise ratio).*

Response 10: We pointed out this finding more clearly in several sections of the paper, most prominently in the following two paragraphs of the Discussion:

"UDQI induce additional inhomogeneities in observational record. The resulting decrease in the signal-to-noise ratio may
35    decrease the performance of statistical data homogenization methods (Domonkos, 2013). This is particularly problematic in

sparse observational networks, where a high number of breakpoints may result in adjustments that deteriorate the temporal consistency of station records (Gubler et al., 2017). In the Central Andean region, UDQI increase the number of statistically detected breakpoints by about 15 % for TX and TN, and by 50 % for PRCP. They also increase the median breakpoint size by 35 to 40 % (TX and TN) and 60 % (RPCP), and increase break size maxima by up to 100 % (temperature) and 70 % (PRCP). Since UDQI have larger relative effects on break sizes than on the number of detected breakpoints, they apparently deteriorate the detectability of small non-climatic inhomogeneities."

"The effect of UDQI also manifests in the adjustment values (temperature) and factors (PRCP) resulting from the data homogenization process. UDQI cause a reduction in the frequency of small adjustments and an increase of large adjustments. They also may induce an adjustment bias. For instance, the median adjustment value for TN station records in the Altiplano is -0.5 °C. If the same dataset contains UDQI, the resulting median adjustment is +0.2 °C. This difference of the adjustment could be caused in two ways. First, UDQI may introduce a systematic bias (a cold bias in earlier observations in this case). This would require the occurrence of certain types of UDQI in many station records of a dataset which would cause a systematic bias and which would strongly change their frequency in time. For the Central Andean area, however, there is no clear indication that UDQI meet these requirements in the period 1981 to 2010. Second, UDQI may not introduce a bias by themselves, but they impede the detection of an existing bias (warm bias in earlier observation in case of TN in the Altiplano) by introducing artificial noise. Such a warm bias could have been introduced, for example, by location changes of weather stations to systematically different sites (e.g. further away from buildings). The second possibility seems to be the more likely cause of the observed adjustment differences of TN records in the Altiplano. Hence, UDQI may impede the adjustments of systematic biases introduced by inhomogeneities. In summary, UDQI may substantially decrease the performance of statistical data homogenization methods."

**Minor comments**

*P2 line 14 – should be 'lose significance'.*

Response 11: Corrected.

*P6 line 30 – 'Spearman' should be capitalised.*

Response 12: Corrected.

*P7 line 9 – 'usually becomes slightly negative' – do you mean 'weaker'? I wouldn't expect a day shift to reduce a strong positive correlation to below 0.*

Response 13: Shifting two highly correlated time series by one day indeed often results in a slightly negative correlation of the first differences in this tropical area. We adapted the relevant sentence as follows:

"For example, a high correlation of two Central Andean time series of the first differences often becomes slightly negative if one of the two time series is shifted by one day."

*P8 line 1 – should be 'performances'*

Response 14: Corrected.

*P10 line 30 – 10.2_C seems very large – is this actually a data quality problem large enough to trigger homogeneity checks?*

Response 15: Data quality issues often trigger the statistical detection of inhomogeneities. This very large adjustment did occur in the time series shown below (Fig. R1) between the segments 1990-1994 and 1997-2007. This time series is (at least temporarily) affected by several systematic data quality issues: strong asymmetric rounding patterns (e.g. around 2000), moderate and strong irregularities in the data patterns (not clearly classifiable but highly suspicious patterns, such as strong changes in the variance, truncations, or missing temperature ranges), and frequent and large inhomogeneities. The low quality of the data suggests that the inhomogeneity rather originates from an observer error (e.g. erroneous reading of the instrument) than from other impacts (e.g. station relocation).

[Figure]

Figure R1. Time series for which the largest temperature adjustment values were calculated in the data homogenization process.

*P11 line 23 – 'inflation of the trend spread' – this doesn't make sense as the trend spread is decreasing, not increasing. I guess you could use 'deflation', but I think just 'decrease' is fine.*

Response 16: Corrected.

*P13 line 13 – should read 'Peruvian Andes and Switzerland'.*

Response 17: Corrected.

*P15 line 5 – should read 'trends of a few climate change indices'*

Response 18: Corrected.

*P15 line 7 – should read 'drawing of clear conclusions'*

Response 19: Corrected.

5    *P15 line 16-18 – while missed observations of a few millimetres may have a negligible effect on monthly sums, they still affect indices which use number of rain days > 1 mm as a basis (e.g. SDII).*

Response 20: Here, we want to stress the fact that UDQI may not have the same effect in different regions. Missing observations of small PRCP events (e.g. up to 2 mm) may not affect the monthly precipitation sums in a wet region, but in a dry area these missed observations may bias monthly PRCP sums. We adapted the relevant sentence as follows:

10   "For instance, missed measurements of small precipitation events of up to a few millimetres may only have a negligible effect on monthly sums in wet regions (e.g. Amazonian Lowlands), whereas they may significantly bias monthly PRCP sums in rather dry areas (e.g. Altiplano) due to low overall PRCP and evaporation losses."

*References – I note that the Gubler et al paper is now published.*

We want to thank the anonymous referee #2 for the important and helpful comments and suggestions.

**Anonymous Referee #2**

5      *General comments*

*This paper uses data from the central Andes to illustrate the impact of systematic data issues on climate analyses. By applying standard and enhanced quality control procedures before data homogenisation, the study shows that systematic biases – which might not be detected using standard QC methods – can have a big impact on the analysis of temperature and rainfall trends.*

10    *This paper is a nice contribution to the field of climate data quality, and raises some important considerations that are often overlooked when large-scale climate trend analysis is conducted. The study is succinct and well written, and is a good complement to this group's previous paper (Hunziker et al. 2017).*

*Some suggestions for possible improvements to the paper are given below for the authors' consideration.*

15    *Specific comments*

*1. The main concern I have about the results from the study is how much data are removed in the enhanced QC procedure, and what impact that would have on the final trends. Around 40% is a significant amount of data, and you'd expect that removing that much information would have an effect on the homogenisation and analyses whether or not the data have issues.*

20    *It'd be great if the authors could repeat their analysis using a synthetic dataset, or a dataset that does not have systematic data issues and see the impact of removing 40% of the data. That way they could ascribe some statistical significance to their findings. If this is not plausible, then at least a discussion on the role of missing data in the results, OR more detail on where and when the removed data are.*

Response 1: We agree with the referee that 40% of the data is a relatively significant amount of information. The effect of
25    removing 40% of the data, however, will strongly differ between different datasets. Furthermore, the results may differ significantly depending on which time series segments are (randomly) excluded. In the authors' opinion, the best approach would be to resample all the available station data quality controlled with the standard QC approach by removing 40% of the data randomly. Repeating this process many times, the resulting climatological analyses from these random groups may provide information about the statistical significance of the results of the present paper. However, the process from the dataset
30    to the climatological analyses is time consuming (clustering, data homogenization, data analyses). Therefore, such an approach is hardly feasible in frame of the present study. Two main lines of evidence make it very unlikely that the differences found in the present study are just an effect of removing 40% of the data: 1) the enhanced QC removes obvious observation errors that cannot be corrected by data homogenization and consequently affect climatological analyses, and 2) the results of the dataset

quality controlled with the enhanced approach are highly consistent (e.g., apparently better performance of data homogenization approach, more consistent station trends, low trend spread of the diurnal temperature range which indicates that the individual data homogenization of TX and TN are in accordance). We will add more information on the availability of time series for the different analyses (for the standard and the enhanced QC approach), and the frequency and temporality of the occurrence of the different data quality issues.

In the revised version of the paper, we included much more information on the spatial and temporal occurrence of data quality issues (see the new Table 1 and Figure 2). We also included the number of available time series (for both QC approaches) for the most important climatological analyses. At the very end of the Discussion, we added the paragraph:

"Removing a relatively large fraction of observations (such as 40 % in the present study) from a dataset may affect the results of climatological analyses. Reducing the spatial density of available data normally decreases the quality of the results such as for data homogenization (Caussinus and Mestre, 2004; Domonkos, 2013; Gubler et al., 2017). With the present study, however, we have demonstrated that removing time series segments affected by UDQI increase the overall quality of the dataset, and results of climatological analyses are consequently more coherent and reliable. The disadvantage of fewer available observation is outperformed by the quality increase of the dataset."

*2. In section 3.1.2, I think a little more information is needed about the QC issues mentioned. I know it's in Hunziker et al. 2017, but the two papers are independent and should be understandable on their own.*

Response 2: In the revised version, we included a new table (Table 2) summarizing and describing all data quality issues considered in the present study.

**Technical corrections**

*• Introduction line 2: replace "most" with "many national"*

Response 3: Adapted.

*• Intro lines 20–30: This part confused me a little. Are you saying that trends actually do vary between neighbouring stations (due to climate factors), or that issues with data quality cause the differences?*

Response 4: The observation of varying trends between neighboring stations was a motivation to investigate if UDQI can cause or increase such inconsistencies. Later in our paper, we show that UDQI indeed may increase spatial trend inconsistencies (Figure 7). We revised the relevant paragraph as follows:

"Trend magnitudes and signs in station records may strongly differ among neighbouring stations. This was observed in many parts of the world and for various climate variables and indices, such as minimum temperature (López-Moreno et al., 2016), precipitation (Rosas et al., 2016; Vuille et al., 2003), diurnal temperature range (Jaswal et al., 2016; New et al., 2006), or extremes indices (Skansi et al., 2013; You et al., 2013). Certain trend differences may be expected even on short spatial scales due to factors such as topography and feedback processes (You et al., 2010). However, errors in observations may also affect individual station trends and hence increase the trend spread within a region."

*• Page 3, line 1: as "the" enhanced approach*

Response 5: Corrected.

*• Page 3, lines 5–9: The Central Andean region is also a good case study because of its complex terrain, as quality issues/homogenisation is notoriously difficult in such conditions.*

Response 6: We added the following sentence to the relevant paragraph:

"Furthermore, the topography in the area is complex, and station density is sparse, making QC and data homogenization difficult."

*• Page 3, lines 9–12: Change chapter to sections*

Response 7: Adapted.

*• Page 3, line 20: add "network" after weather observation*

Response 8: Corrected.

*• Page 4, line 5: data "were" quality-controlled*

Response 9: Corrected.

*• Page 4, lines 10–15: I'd add that the Durre et al. tests include spatial inconsistency tests.*

Response 10: Included.

*• Page 4, lines 27–29: Why doesn't the GHCN-Daily QC approach flag these clearly erroneous values?*

Response 11: The reason for the problem is that some extremely high and low numbers did not fit in the maximum allowable number of digits in the GHCN-Daily data format. We adapted the relevant paragraph as follows:

"Furthermore, the GHCN-Daily QC did not flag a few extreme outliers. This may happen if a reported value exceeds the maximum of five places in tenth of °C or mm allowed in the GHCN-Daily data format (e.g. values $\leq$-10000). In order to remove such erroneous numbers, we added an additional flag to all unflagged temperature values >70 °C and < 70 °C, as well as to all unflagged negative PRCP values."

*• Page 4, line 30: add % after 0.35*

Response 12: Corrected.

*• Page 5, line 6: on "an" annual time scale*

Response 13: Corrected.

*• Page 5, line 26: add % after 0.26*

Response 14: Corrected.

*• Page 5, lines 28 and 30 [and throughout the manuscript]: I think you need "a" before monthly and daily time scale.*

Response 15: Corrected.

5 *• Page 6, line 5–6: I suggest you reword to: Note that Hunziker et al. (2017) further suggest the inclusion of additional information derived from 5 metadata into the QC process. This allows the removal station records that were generated under inappropriate conditions such as poor station siting or severe…*

Response 16: Adapted.

*• Page 6, line 15: one day "a week"*

10 Response 17: Corrected.

*• Page 6, line 30: I'd add reference to Peterson et al. 1998 when discussing using firstdifferences: Peterson, T. C., T. R. Karl, P. F. Jamason, R. Knight, and D. R. Easterling (1998), First difference method: Maximizing station density for the calculation of long-term global temperature change, J. Geophys. Res., 103(D20), 25967–25974, doi:10.1029/98JD01168.*

15 Response 18: We did not include the suggested reference, because the First difference method described by Peterson et al. is not exactly what we did. However, we revised the relevant sentence in order to make it clearer:

"In order to remove the influence of trends and inhomogeneities, the differences between one observation and the next were calculated. From these time series of the first differences, Spearman rank correlations were computed for the period 1981 to 2010."

20 *• Page 6, line 31: Spearman (with a capital S)*

Response 19: Corrected.

*• Page 7, line 1: add "the" before monthly time scale*

Response 20: Corrected.

*• Section 3.4.1. Is there a way to graphically show the different clusters? Perhaps some additional panels in Figure*
25 *1 using polygons?*

Response 21: The clusters differ for each parameter and for the different QC approaches. Therefore, they cannot be marked in Figure 1. We included a summarizing figure (Figure S1) showing maps of the different clusters in the supplementary material.

*• Section 3.4.2. It sounds like you've used ACMANT3 because it is fully automated. Perhaps mention this earlier in the section to make that point stronger.*

Response 22: We adapted the relevant paragraph as follows:

"There are various established homogenization approaches (e.g. Aguilar et al., 2003; Ribeiro et al., 2016; Venema et al., 2012). For the present study, the method ACMANT was chosen. ACMANT is a fully automatic method that does not incorporate metadata. Hence, the approach is objective, in contrast to semi-automatic approaches such as HOMER (Mestre et al., 2013) that require various subjective decisions. This subjectivity may influence the results of the homogenization process (Vertačnik et al., 2015). For the aim of the present study to evaluate the effects of undetected data quality issues, it is important to avoid such disturbances. ACMANT is a state of the art homogenization method having one of the best performances (Ribeiro et al., 2016; Venema et al., 2012). Recently, a new version of the approach (ACMANT3) was published (Domonkos and Coll, 2017). Compared to previous versions (Domonkos, 2011; Domonkos, 2015), the performance of the method was further improved and the range of use increased (Domonkos and Coll, 2017)."

*• Page 8, line 14: remove "of"*

Response 23: Corrected.

*• Page 8, line 15: remove the hyphen between DECADE and dataset.*

Response 24: Corrected.

*• Page 8, line 16: add "and" after requirements*

Response 25: Corrected.

*• Page 8, lines 20–24: You need an extra line or two here to define/explain the Theil-Sen estimator, and how you have pre-whitened the data.*

Response 26: We will include the required information.

*• Page 9, lines 5-15: I understand what you're trying to say here, but it's a bit confusing. Consider revising the text.*

Response 27: We adapted the relevant paragraph as follows:

"Magnitudes of linear trends were calculated with the Theil-Sen estimator, which is calculated by the median of the slopes of all data pairs of a time series (Sen, 1968; Theil, 1950). The method is more insensitive to outliers and more robust than other trend estimators such as Ordinary Least Squares. For individual station records, the significance of trends is not of major interest in the present study and was therefore not tested. Furthermore, taking serial correlation into account in trend tests would cause large uncertainties due to the missing values in the time series. However, for the Altiplano stations, trends of spatially averaged anomalies were tested with the Mann-Kendall test at the 5 % significance level. Before applying the Mann-Kendall test, the time series were pre-whitened (Frei, 2013) in order to remove the influence of serial correlation."

*• Page 13, line 13: "and" Switzerland*

Response 28: Corrected.

> *• Page 13, line 14: Why are you hypothesising this? Why would the tropics have more UDQI? Do you also mean that there are more likely to be UDQI in developing countries?*

Response 29: Weather observation in less developed countries is at a different stage than weather observation in industrialized countries. Therefore, it is likely that more data quality issues occur in data from less developed countries that are particularly located in the tropics. We adapted the relevant sentences as follows:

"Hypothesising that UDQI occur more frequently in station networks of developing than developed countries, a higher frequency of such errors can be expected in tropical areas than in mid-latitudes. Making this assumption, UDQI may partly explain the particularly low correlation decay distances in the tropics described by New et al. (2000)."

> *• Page 15, line 5: add "a" before few climate change indices*

Response 30: Corrected.

> *• Table 1 caption: were, not where*

Response 31: Corrected.

> *• Table 2: can you add any information about the spread of the data in this table?*

Response 32: We think that adding the spread of the data to the table would not add much information. The spread will be highly dependent on the topography and on the availability of data. We tried to get the best possible estimate of the median correlation by adding the shifting of daily values and finding the highest correlation (accounting for day shifts in the data). If in some cases, very low or negative correlation are erroneously altered by this approach, it will not have a significant effect on the median, but it may bias the spread. Therefore, we prefer to not include this information in the table.

> *• Figure 2 caption: I'd add the word "show" after the words green triangles and red triangles.*

Response 32: Corrected.

[revised manuscript text omitted]
\text{-}S}$ | DATA$_{QC\text{-}S\ H}$ | DATA$_{QC\text{-}E}$ | DATA$_{QC\text{-}E\ H}$ | DATA$_{QC\text{-}S}$ | DATA$_{QC\text{-}S\ H}$ | DATA$_{QC\text{-}E}$ | DATA$_{QC\text{-}E\ H}$ |
| Annual means (°C decade$^{-1}$) | **+0.41** | **+0.42** | **+0.44** | **+0.40** | -0.04 | +0.05 | -0.12 | **+0.22** |
| 10[th] percentile (days decade$^{-1}$) | **-13.2** | **-14.4** | **-12.0** | **-11.9** | +0.4 | - 1.0 | -0.9 | **-5.8** |
| 90[th] percentile (days decade$^{-1}$) | **+8.7** | **+11.0** | **+9.8** | **+9.3** | +0.2 | +3.7 | +2.0 | **+8.8** |
| FD (days decade$^{-1}$) | --- | --- | --- | --- | +2.9 | -1.3 | +1.4 | **-6.5** |

[revised manuscript text omitted]

---

## Author Response (AR2)

We thank the anonymous referee #2 for carefully reviewing the manuscript and for helping to avoid errors and lacks of clarity.

**Response to Anonymous Referee #2**

5 *Page 3, line 13: 'de' should be corrected to 'the'*

Corrected

*Page 5, line 31, and throughout: you have used data as singular here (i.e. 'data was', rather than 'data were'), but as a plural elsewhere. Please check the manuscrip one last time to make sure you are consistent*

Corrected

10 *Page 9, line 30: swap 'varies' and 'clearly'*

Corrected

*Page 9, line 32: 'personnel at the airports ARE'*

Corrected

*Page 10, line 5: 'but barely occur', rather than 'but do barely occur'*

15 Corrected

*Page 10, line 7: 'hence this issue is largely absent'*

Corrected

*Page 10, line 27: 'on a monthly and daily scale' (remove the 'a' before daily)*

Corrected

20 *Page 11, line 25: 'approach' not 'approache'*

Corrected

*Page 11, line 28: 'detected on average'*

Corrected

> *Page 12, line 8: 'reference stations', rather than 'references'*

Corrected

> *Page 17, line 32: what do you mean by 'cannot take UDQI into account'? I think you mean that UDQI must be considered and ideally removed, but it is not clear in this statement*

5    We adapted the sentence as follows:

"Hence, data homogenization methods rely on data that are largely free of UDQI in order to perform satisfactorily."

> *Table 1 caption: I would refer to Figure 1 here when mentioning the different regions, and also add a sentence at the end pointing readers to Hunziker et al 2017 for extra information on the tests.*

The caption was adapted as suggested.

[revised manuscript text omitted]